# Titans: Learning to Memorize at Test Time

**Ali Behrouz**
Google Research
USA
alibehrouz@google.com

**Peilin Zhong**
Google Research
USA
peilinz@google.com

**Vahab Mirrokni**
Google Research
USA
mirrokni@google.com

## Abstract

Over more than a decade there has been an extensive research effort on how to effectively utilize recurrent models and attention. While recurrent models aim to compress the data into a fixed-size memory (called hidden state), attention allows attending to the entire context window, capturing the direct dependencies of all tokens. This more accurate modeling of dependencies, however, comes with a quadratic cost, limiting the model to a fixed-length context. We present a neural long-term memory module that learns to memorize historical context and helps attention to attend to the current context while utilizing long-past information. We show that this neural memory has the advantage of fast parallelizable training. From a memory perspective, we argue that attention due to its limited context but accurate dependency modeling performs as a short-term memory, while neural memory due to its ability to memorize the data, acts as a long-term, more persistent, memory. Based on these two modules, we introduce a new family of architectures, called Titans, and present three variants to address how one can effectively incorporate memory into this architecture. Our experimental results on language modeling, common-sense reasoning, and time series tasks show that Titans are effective compared to baselines, while they can effectively scale to larger context window in needle-in-haystack tasks.

## 1 Introduction

Transformers [1], have been firmly established as state-of-the-art models in sequence modeling, mainly due to their in-context learning and ability to learn at scale [2]. The primary building blocks of Transformers–attention modules—function as associative memory blocks [3], where they learn to store key-value associations and retrieve them by computing pairwise similarity between queries (i.e., search signals) and keys (i.e., contexts). Accordingly, by design, the output of a Transformer is exclusively conditioned on the direct dependencies of tokens in the *current* context window. This accurate modeling of dependencies, however, comes with quadratic time and memory complexity in terms of the context length. In complex real-world tasks (e.g., language modeling [4], video understanding [5], long-term time series forecasting [6]), the context window can become extremely large, making the applicability of Transformers challenging in these downstream tasks.

To overcome the scalability issue of Transformers, recent studies aim to design different variants of linear Transformers [7–9], where softmax is replaced by a kernel function in the attention (see §A.1 for details), resulting in a significant drop in memory consumption. Despite efficiency and the ability to scale to longer context, linear Transformers do not show competitive performance compared to Transformers as the kernel trick makes the model a linear recurrent network, in which the data is compressed into a matrix-valued states [7]. This, however, brings a contradictory fact about linear recurrent (or linear Transformers) models: On one hand, we use these linear models to enhance scalability and efficiency (linear vs. quadratic complexity), whose advantages is appeared

39th Conference on Neural Information Processing Systems (NeurIPS 2025).

for very long context; On the other hand, a very long context cannot be properly compressed in a small vector-valued or matrix-valued states [10].

Furthermore, beyond efficiency, most existing architectures–ranging from Hopfield Networks [11] to LSTMs [12] and Transformers [1]–face challenges when dealing with generalization, length extrapolation, and/or reasoning [13, 14], all of which are inseparable parts of many hard real-world tasks. Although these architectures draw inspiration from the human brain, each of which are missing: (1) a crucial component for learning process—such as short-term memory, long-term memory, meta-memory, attending to current context, etc. [15]; (2) how these components are interconnected systems that can operate independently; and/or (3) the ability to *actively* learn from data and memorize the abstraction of past history. We argue that in an effective learning paradigm, similar to human brain, there are *distinct* yet interconnected modules, each of which is responsible for a component crucial to the learning process.

**Memory Perspective.** Memory is a fundamental mental process and is an inseparable component of human learning [16]. Without a properly functioning memory system, humans and animals would be restricted to basic reflexes and stereotyped behaviors. Accordingly, memory has been the inspiration for many seminal research in machine learning literature; e.g., Hopfield Networks [11], LSTMs [12], and Transformers [1].

Taking inspiration from the common definitions of memory and learning in neuropsychology literature [17], most existing architectures consider memory as a neural update caused by an input, and define learning as a process for acquiring effective and useful memory, given an objective. In this perspective, Recurrent Neural Networks (RNNs) [18] can be defined as models with a vector-valued memory module $\mathcal{M}$ (also called hidden state) with two main steps: Given a new input $x_t$ at time $t$, the model (1) updates the memory using a function $f(\mathcal{M}_{t-1}, x_t)$ (with compression); and (2) retrieves the corresponding memory of input using a function $g(\mathcal{M}_t, x_t)$. Similarly, Transformers can be seen as architectures with a growing memory and two similar steps. That is, the pair of key and value matrices acts as the model's memory, and the model: (1) updates the memory by appending the key and value to the memory (without compression), and (2) retrieves query vectors' corresponding memory by finding the similarity of query and key vectors, which is then used to weight the value vectors for the output.

This perspective, can help us better understand existing paradigms, their critical differences, and design more effective architectures. For example, the main difference between Transformers [1] and *linear* Transformers [7] is the memory structure as well as the memory updating step, in which linear Transformers compress the historical data into a fixed-size matrix-valued memory while Transformers keep all historical data (within the context length) without any compression. Therefore, this perspective motivates us to ask: **(Q1)** What is a proper memory update mechanism? and **(Q2)** What is a good memory retrieval process?

Human memory is neither a unitary process nor it serves a single function [15]. In fact, memory is a confederation of systems–e.g., short-term, working, and long-term memory–each serving a different function with different neural structures, and each capable of operating independently [19]. This motivates us to ask: **(Q3)** How to design an efficient architecture that incorporates different interconnected memory modules. Finally, storing a memory is a neural process that requires to encode and store the abstraction of the past. It can be over-simplification to assume a single vector or a matrix, whose parameters are encoding the data in a linear manner, are enough for storing long-term history. **(Q4)** Is a deep memory module needed to effectively store/remember long past?

**Neural Memory** (§2). We present a (deep) neural long-term memory that (as a meta in-context model) learns how to memorize/store the data into its parameters at test time. Inspired by human long-term memory system [20], we design this memory module so an event that violates the expectations is more memorable. To better handle the limited memory, we present a decaying mechanism that consider the proportion of memory size and the amount of data surprise, resulting in better memory management.

**Titans Architectures** (§3). After designing the long-term neural memory, an important remaining question is how to effectively incorporate memory into an architecture. We present Titans, a family of deep models that consists of three hyper-heads: (1) Core: this module consists of the short-term memory, and is responsible for the main flow of processing the data (we use attention with limited window size); (2) Long-term Memory: this branch is our neural long-term memory module that

is responsible to store/remember long past; (3) Persistent Memory: this is a set of learnable but data-independent parameters that encodes the knowledge about a task. Finally, as a proof of concept, we present three variants of Titans, in which we incorporate memory as: (i) a context, (ii) a layer, and (iii) a gated branch.

We provide an extensive list of related work and background concepts in Appendix A and Appendix B.

## 2   Learning to Memorize at Test Time

To overcome the lack of long-term memory and to enable the model to learn, forget, and retrieve information, in this section, inspired by the concept of Test Time Training [21–23], we present a neural long-term memory module, which is a meta models that learns to memorize at test time. In Section 2.1, we first discuss the motivation and the design of the neural memory. In Section 2.2, we discuss how our architecture design can benefit from a fast and parallelizable training. Finally, in Section 2.3, we augment our architecture using persistent memory module, in which we use learnable but data-independent parameters to learn meta information about the task.

### 2.1   Long-term Memory

To design a neural long-term memory, we need a model that can encode the abstraction of the past history into its parameters. An example of this can be deep large models that are shown to be memorizing their training data [24–26]. Therefore, a simple idea is to train a neural network and expect it to memorize its training data. Memorization, however, has almost always been known as an undesirable phenomena in neural networks as it limits the model generalization [27], causes privacy concerns [24], and so results in poor performance at test time. Moreover, the memorization of the training data might not be helpful at test time, in which the data might be out-of-distribution. We argue that, we need an online meta-model that learns how to memorize/forget the data at test time. In this setup, the model is learning a function that is capable of memorization, but it is not overfitting to the training data, resulting in a better generalization at test time.

**Learning Process and Surprise Metric.** The key idea to train a long-term memory is to treat its training as an online learning problem, in which we aim to compress the past information $x_1, \ldots, x_{t-1}$ into the parameters of our long-term neural memory module $\mathcal{M}_t$. As discussed earlier, an event that violates the expectations (i.e., is surprising) is more memorable for humans [20]. Inspired by this, a simple definition of surprise for a model can be its gradient with respect to the input. The larger the gradient is, the more different the input data is from the past data. Accordingly, using this surprise score, we can update the memory as $\mathcal{M}_t = \mathcal{M}_{t-1} - \theta_t \nabla \ell(\mathcal{M}_{t-1}; x_t)$. This surprise metric, however, can result in missing important information that comes after a big surprising moment. That is, the gradient can become extremely small after several surprising steps, leading to stocking in a flat area (i.e., local minima), and missing information about some parts of the sequence. From the human memory perspective, an event might not consistently surprise us through a long-period of time although it is memorable. The reason is that the initial moment is surprising enough to get our attention through a long time frame, leading to memorizing the entire time frame. To improve the above surprise metric, we break the surprise metric into (1) *past surprise*, which measures the surprise amount of a very recent past; and (2) *momentary surprise*, which measures the surprise of incoming data:

$$\mathcal{M}_t = \mathcal{M}_{t-1} + S_t, \qquad S_t = \eta_t \underbrace{S_{t-1}}_{\text{Past Surprise}} - \theta_t \underbrace{\nabla \ell(\mathcal{M}_{t-1}; x_t)}_{\text{Momentary Surprise}}. \tag{1}$$

Interestingly, this formulation is similar to gradient descent with momentum, where $S_t$ is the momentum element. Therefore, the momentum here act as a memory of surprise across time (sequence length). In this formulation, the term $\eta_t$ is a data-dependent surprise decay (a function of $x_t$), controlling how surprise decays over time, and the term $\theta_t$ is controlling how much of momentary surprise should be incorporated into the final surprise metric in a data-dependent manner. This data-dependency is particularly important in this design: While surprise of previous tokens might be needed to affect the surprise of the next token, it is mostly valid if all tokens are relevant and are in the same context. Accordingly, a data-dependent $\eta$ can control if memory needs to: (1) ignore the last surprise by setting $\eta_t \to 0$ (possibly due to the change of context), or (2) fully incorporate the last surprise by setting $\eta_t \to 1$ (possibly as the token is highly relevant to its recent past tokens).

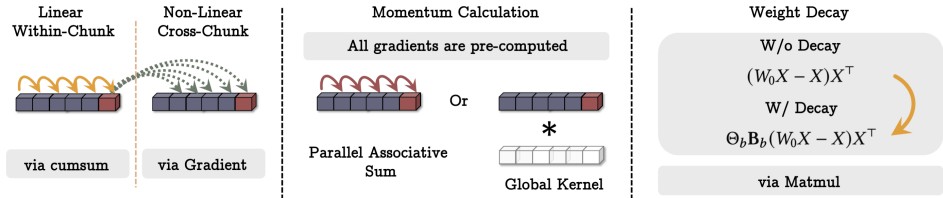

Figure 1: The illustration of how neural memory's training can be done in parallel and using `matmuls`.

**Objective.** Our above surprise metric is based on a loss function $\ell(.;.)$, which is the objective that our memory is learning to act as it at test time. That is, our memory module is a meta model that learns a function based on the loss function $\ell(.;.)$. In this work, we focus on *associative memory*, in which we aim to store the past data as the pairs of keys and values. Given $x_t$, similar to Transformers, we use linear layers to project $x_t$ into a key and value: $\mathbf{k}_t = x_t W_K$, and $\mathbf{v}_t = x_t W_V$, where $W_K$ and $W_V \in \mathbb{R}^{d_{in} \times d_{in}}$. Next, we expect our memory module to learn the associations between keys and values. To this end, we define the loss as follows:

$$\ell(\mathcal{M}_{t-1}; x_t) = \|\mathcal{M}_{t-1}(\mathbf{k}_t) - \mathbf{v}_t\|_2^2 \tag{2}$$

By optimizing the above loss function in the inner-loop of our meta model (memory), the model learns how to memorize the mapping between keys and values at test time. Note that, similar to meta-learning models [28, 29], training of the memory is in the inner-loop, and so parameters $W_K$ and $W_V$ are hyperparameters in the above loss function. Accordingly, in the inner loop, we optimize $\mathcal{M}$'s weights, while in the outer-loop, we optimize other parameters of the entire architecture.

**Forgetting Mechanism.** When dealing with very large sequences, it is crucial to manage which past information should be forgotten–even with a deep or a very large matrix-valued memory. To this end, we use an adaptive forgetting mechanism that allows the memory to forget the information that is not needed anymore, resulting in better managing the memory's limited capacity. That is, given the next token $x_t$, we modify the update rule as:

$$\mathcal{M}_t = (1 - \alpha_t)\mathcal{M}_{t-1} + S_t, \qquad S_t = \eta_t S_{t-1} - \theta_t \nabla \ell(\mathcal{M}_{t-1}; x_t), \tag{3}$$

where $\alpha_t \in [0, 1]^{d_{in}}$ is the gating mechanism that flexibly controls the memory; i.e., decides how much information should be forgotten. For example, it can update the memory without affecting the past abstraction by letting $\alpha_t \to 0$, and can clear the entire memory by letting $\alpha_t \to 1$. Later in this section, we show that this weight decay mechanism is closely related to the gating mechanism in modern RNNs [30, 31].

**Memory Architecture.** In this paper, we focus on simple MLPs with $L_{\mathcal{M}} \geq 1$ layers as the architecture of our long-term memory. The main reason behind this choice is that we want to focus on better motivating the design of the long-term memory and ways that it can be incorporated into an architecture. Recently, there has been a promising line of work to design architectures that are better memorizers [32–34]; incorporating them into our framework (i.e., replacing simple MLPs with such architectures) can be an interesting future work.

**Retrieving a Memory.** In the above, we discuss how one can design and train a long-term memory module that learns to memorize at test time. A key remaining question is: *How one can retrieve information from the memory?* We simply use the forward pass without weight update (i.e., inference) to retrieve a memory correspond to a query. Formally, given an input $x_t$, we use a linear layer $W_Q$ to project the input, i.e., $\mathbf{q}_t = x_t W_Q$ and retrieve the corresponding information from the memory $y_t$ by $y_t = \mathcal{M}(\mathbf{q}_t)$.

### 2.2 How to Parallelize the Long-term Memory Training

As discussed above, the design of our long-term memory module is equivalent to training a meta model by optimizing associative memory loss function $\ell(\mathcal{M}_{t-1}; x_t) = \|\mathcal{M}_{t-1}(\mathbf{k}_t) - \mathbf{v}_t\|_2^2$ using gradient descent with momentum and weight decay. Therefore, in theory, the training of long-term memory module requires $\mathcal{O}(N)$ FLOPs, where $N$ is the sequence length. However, in practice, we

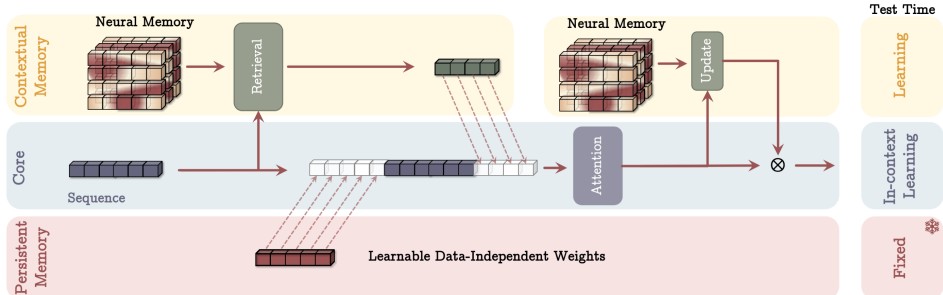

Figure 2: **Memory as Context (MAC) Architecture.** This architecture includes three branches of (1) core, (2) contextual (long-term) memory, and (3) persistent memory. The core branch concatenates the *corresponding* long-term and persistent memories with the input sequence. Attention then performs on the sequence and decides what part of the information should be stored in the memory.

need to parallelize the training process and to fully take advantage of hardware accelerators (e.g., TPUs, GPUs), we need to tensorize the process and use more `matmuls`.

Similar to Sun et al. [21] and contrary to recent modern linear recurrent models [35, 36, 30], our formulation of neural memory is not a linear recurrence. Therefore, it cannot be written as an associative operator, and so be trained by parallel scan [37]. Accordingly, we use the idea of chunk-wise recurrence [38, 21], and instead of $\nabla\ell(\mathcal{M}_{t-1}; x_t)$, we use $\nabla\ell(\mathcal{M}_{t'}; x_t)$ in Equation 3, where $t'$ is the last step in the previous chunk (see Figure 1).

To fully take advantage of accelerators, we need to use `matmuls` and sum. We build upon the work of Sun et al. [21] that shows forward pass of a model optimizing with the mini-batch gradient descent (with constant learning rate) can be calculated using `matmuls`. As discussed above, we split the sequence into chunks of size $b \geq 1$, and write the mini-batch gradient descent as:

$$\mathcal{M}_t = (1 - \alpha_t)\mathcal{M}_{t-1} - \theta_t\nabla\ell(\mathcal{M}_{t-1}; x_t) = \beta_t\mathcal{M}_0 - \sum_{i=1}^{t} \theta_i\frac{\beta_t}{\beta_i}\nabla\ell(\mathcal{M}_{t'}; x_i),$$

where $t' = t - \text{mod}(t, b)$, and $\beta_i = \prod_{j=1}^{i}(1 - \alpha_j)$. For the sake of simplicity, we focus on the first chunk, i.e., $t = b$ and so $t' = 0$. Also, we explain the process for the case that $\mathcal{M}_t = W_t$ is linear. The process for MLPs with $N_p \geq 2$ is similar. Using our loss function, we have:

$$\nabla\ell(W_0; x_t) = (W_0 k_t - v_t)x_t^\top \Rightarrow \sum_{i=1}^{b} \theta_i\frac{\beta_b}{\beta_i}\nabla\ell(W_0; x_i) = \Theta_b \mathbf{B}_b(W_0 K - V)X^\top, \qquad (4)$$

where $\Theta_b = \text{diag}\left(\begin{bmatrix} \theta_1 & \theta_2 & \dots & \theta_b \end{bmatrix}\right)$ and $\mathbf{B}_b$ is defined analogously on $\frac{\beta_b}{\beta_i}$s. Note that, we do not need to store all $\Theta_{kb}$ and $\mathbf{B}_{kb}$ for $k = 1, \dots, N/b$, instead, we store these matrices for each chunk, and so using less memory. Next, we extend this representation so we can incorporate the momentum term. If we look at the momentum term, we have:

$$S_t = \eta_t S_{t-1} - \theta_t u_t, \qquad (5)$$

where $u_t = \nabla\ell(\mathcal{M}_{t'}; x_t)$. Note that, we can compute all $u_t$ at the same time, and so Equation 5 is a linear recurrence with $u_t$s as inputs, $S_t$ as the state, and $\eta_t$ as input-dependent transition value. Accordingly, we can use parallel associative scan [37] to calculate $S_t$s in each chunk.

## 2.3 Persistent Memory

Our long-term memory can also be seen as a contextual memory, meaning that the output is fully depend on the context. Therefore, in addition to our long-term memory, we also use a set of learnable but input-independent parameters to act as task-related memory. This type of memory has been referred to as persistent or meta-memory in the literature [39, 40]. Given $N_p \geq 1$, we use learnable parameters $P = \begin{bmatrix} p_1 & p_2 & \dots & p_{N_p} \end{bmatrix}$ and append it to the start of our sequence: i.e., given a context window size of $N$, we modify the input as:

$$x_{\text{new}} = \begin{bmatrix} p_1 & p_2 & \dots & p_{N_p} \end{bmatrix} \,||\, x, \qquad (6)$$

where $||$ is concatenation. We discuss the other motivations for persistent memory in Appendix C.

# 3 How to Incorporate Memory?

An important question that remained unanswered is: How one can effectively and efficiently incorporate the designed neural memory into a deep learning architecture? In this section, we aim to answer the above question by proposing three different variants of Titans. Later in our experiments, we show that each of these variants has its own (dis)advantages and also can show a trade-off between the efficiency and effectiveness in very long-contexts.

## 3.1 Memory as a Context

In the first architecture design (see Figure 2), we treat the memory as a context to the current information. That is, given a long sequence $x \in \mathbb{R}^{N \times d_{\text{in}}}$, we first divide the sequence into fixed-size segments $\mathbf{S}^{(i)}$ for $i = 1, \ldots, N/C$. Given the incoming segment $\mathbf{S}^{(t)}$, we consider it as the current context and its past segment as the historical information. Therefore, let $\mathcal{M}_{t-1}$ be the state of long-term memory before segment $\mathbf{S}^{(t)}$, we use the input context as the query to the memory $\mathcal{M}_{t-1}$ to retrieve the corresponding information from the long-term memory. That is, we retrieve the past information that corresponds to $\mathbf{S}^{(t)}$ as $h_t = \mathcal{M}_{t-1}(\mathbf{q}_t)$, where $\mathbf{q}_t = \mathbf{S}^{(t)} W_Q$. Next, we use this historical information along with our persistent memory parameters as the input sequence to the attention module:

$$\tilde{\mathbf{S}}^{(t)} = \begin{bmatrix} p_1 & p_2 & \cdots & p_{N_p} \end{bmatrix} \; || \; \mathbf{S}^{(t)} \; || \; h_t, \tag{7}$$

$$y_t = \texttt{Attn}\left(\tilde{\mathbf{S}}^{(t)}\right). \tag{8}$$

The structure of the attention map over the entire sequence is shown in Figure 4a. We then use $y_t$ to update the long-term memory module for the next segment and the final output:

$$\mathcal{M}_t = \mathcal{M}_{t-1}\left(y_t\right), \tag{9}$$

$$o_t = y_t \otimes \mathcal{M}_t\left(y_t\right). \tag{10}$$

This architecture has two key advantages: (1) Attention by having both historical and current context, has the ability to decides whether given the current data, the long-term memory information is needed. (2) The attention module helps the long-term memory to store only useful information from the current context. That is, not all tokens in each segment are useful and memorizing all of them can result in memory overflow. Therefore, attention is helping the memory to understand which information is useful, better managing the memory capacity. At test time: (i) persistent memory parameters are fixed as they encodes the knowledge about the task, which should not be changed; (ii) the attention module weights are in-context learner; and (iii) the long-term memory module is still learning (memorizing) the information at test time.

It is notable that we use chunk and segment for two different concepts. Chunks are subsequence that we use to accelerate the training process of the memory. In fact, for each chunk, we take the gradient with respect to the last state of the previous chunk. On the other hand, we use segment to refer to the larger subsequence that we perform attention on (as discussed above).

## 3.2 Gated Memory

In the next variant (see Figure 5), in one branch, we directly use the long-term memory, and in the second branch, we use a sliding window attention (SWA):

$$\tilde{x} = \begin{bmatrix} p_1 & p_2 & \cdots & p_{N_p} \end{bmatrix} \; || \; x, \tag{11}$$

$$y = \texttt{SW-Attn}^*\left(\tilde{x}\right), \tag{12}$$

$$o = y \otimes \mathcal{M}(\tilde{x}), \tag{13}$$

where $\texttt{SW-Attn}^*$ is sliding window attention with prefix (see Figure 4b). Note that, contrary to the previous design, we are not segmenting the input data. Also, we abuse the notation and use $\mathcal{M}(x)$ to refer to the final output of the memory after all recursion over the tokens of the sequence. In the above equation, $\otimes$ can be any non-linear gating. In our experiments, we normalize the outputs $y$ and $\mathcal{M}(\tilde{x})$ using learnable vector-valued weights, followed by a non-linearity. We provide additional motivations/interpretations for this design as well as the visualization of its attention mask in Appendix D.

The last variant is Memory as Layer (MAL), in which we sequentially use neural memory module and attention as the layer of the model. In our experiments, we use a similar architecture as Samba [41], where we replace the the sequence model with our neural memory module (LMM). Additional details are provided in Appendix D.

### 3.3 Architectural Details

**Memory Architecture.** For the memory architecture, we use an MLP with $\mathcal{L}_{\mathcal{M}}$ layers (default is $\mathcal{L}_{\mathcal{M}} = 2$) with expansion factor of 4 and GELU activation function [42]. We also use residual connections and layer norm: $\mathcal{M}(x) = x + \text{LN}(W_1\sigma(W_2x))$.

**Token Mixer.** We follow previous studies [38, 35], and replace the attention with Titans in Llama's macro architecture with MLPs with $\text{SwiGLU}(.)$ activation, rotary positional encodings (RoPE) [43], and RMSNorm [44]. Following the recent modern linear recurrent models [35, 45], we incorporate a 1D convolution layer with size 4 after each of the query, key, and value projections. For the sake of training stability, we also use $\ell_2$ normalization to $q$ and $k$. We also follow the recent architectures that use normalization and gating with a linear layer before the final output projection [46].

Additional details about the architecture and dimensions of models are in Appendix E.

## 4 Experiments

Next, we evaluate the performance of Titans and its variants in language modeling, commonsense reasoning, needle in haystack, DNA modeling, and time series forecasting tasks. In more details, in this section, we answer the following empirical questions: (1) How do Titans perform compared to baselines in downstream tasks? (see §4.2, §G.5, and §G.6); (2) What is the actual context length of Titans? (see §4.3 and §G.1); (3) How do Titans scale with respect to context length? (see §G.4); (4) How the depth of memory can affect both performance and efficiency? (see §4.4); and (5) What is the contribution of each Titans' component in its performance? (see §4.5).

### 4.1 Experimental Setup

**Models.** In our experiments, we focus on the four variants of Titans, which we refer to as: Titans with (1) Memory as a Context (MAC), (2) Memory as a Gate (MAG), and (3) Memory as a Layer (MAL) as well as (4) neural memory module alone. For each of these models, we consider four scales with: (i) 170M, (ii) 340M, (iii) 400M, (iv) 760M, and (v) 1.3B parameters. While the first three are trained on 15B tokens sampled from FineWeb-Edu dataset [47], the last two are trained on 30B and 100B tokens from the same dataset.

**Baselines.** We compare our models with the state-of-the-art linear recurrent models, Transformers, and hybrid models (recurrent + attention). More specifically in language tasks, we compare with Transformer++ [48], RetNet [49], Gated Linear Attention (GLA) [9], Mamba [45], Mamba2 [30], DeltaNet [38], TTT [21], and Gated DeltaNet [35]. In needle in haystack tasks, we also compare with GPT4 [50], Llama3 with RAG [48], RecurrentGemma2-9B [51], and Mistral [52] models, all of which are provided in the benchmark [53]. In time series tasks, we compare with Mamba-based [54], Transformer-based [55–57], and linear models [58–61].

**Training.** In the training, we follow the training procedure of Yang et al. [35], and use LLama 2 tokenizer with a vocabulary size of 32K and use training length of 4K tokens (2K for SWA). We fixed the persistent memory size (# tokens) to 128, and use 256 memory tokens to encode the past data (i.e., output of the long-term memory). We employ AdamW optimizer with learning rate of $4e\text{-}4$ with cosine annealing schedule with batch size of 0.5M tokens, and weight decay of 0.1.

### 4.2 Language Modeling

We first focus on the perplexity in language modeling and also commonsense reasoning tasks. The results for Titans' variants and also baselines with three different sizes of 340M, 400M, and 760M are reported in Table 1 (Table 6). Among non-hybrid models, including Transformer++, our neural

Table 1: Performance of Titans and recurrent- and Transformer-based baselines on language modeling and common-sense reasoning tasks. Hybrid models are marked with *. The best results among simple and hybrid models are highlighted.

| Model | Wiki. ppl↓ | LMB. ppl↓ | LMB. acc↑ | PIQA acc↑ | Hella. acc_n↑ | Wino. acc↑ | ARC-e acc↑ | ARC-c acc_n↑ | SIQA acc↑ | BoolQ acc↑ | Avg. ↑ |
|---|---|---|---|---|---|---|---|---|---|---|---|
| | | | | 760M params / 30B tokens | | | | | | | |
| Transformer++ | 25.21 | 27.64 | 35.78 | 66.92 | 42.19 | 51.95 | 60.38 | 32.46 | 39.51 | 60.37 | 48.69 |
| Mamba | 28.12 | 23.96 | 32.80 | 66.04 | 39.15 | 52.38 | 61.49 | 30.34 | 37.96 | 57.62 | 47.22 |
| DeltaNet | 24.37 | 24.60 | 37.06 | 66.93 | 41.98 | 50.65 | 64.87 | 31.39 | 39.88 | 59.02 | 48.97 |
| TTT | 24.17 | 23.51 | 34.74 | 67.25 | 43.92 | 50.99 | 64.53 | 33.81 | 40.16 | 59.58 | 47.32 |
| Gated DeltaNet | 21.18 | 22.09 | 35.54 | 68.01 | 44.95 | 50.73 | 66.87 | 33.09 | 39.21 | 59.14 | 49.69 |
| Samba* | 20.63 | 22.71 | 39.72 | 69.19 | 47.35 | 52.01 | 66.92 | 33.20 | 38.98 | 61.24 | 51.08 |
| Gated DeltaNet-H2* | 19.88 | 20.83 | 39.18 | 68.95 | 48.22 | 52.57 | 67.01 | 35.49 | 39.39 | 61.11 | 51.49 |
| Titans (LMM) | 20.04 | 21.96 | 37.40 | 69.28 | 48.46 | 52.27 | 66.31 | 35.84 | 40.13 | 62.76 | 51.56 |
| Titans (MAC) | 19.93 | 20.12 | 39.62 | 70.46 | 49.01 | 53.18 | 67.86 | 36.01 | 41.87 | 62.05 | 52.51 |
| Titans (MAG) | 18.61 | 19.86 | 40.98 | 70.25 | 48.94 | 52.89 | 68.23 | 36.19 | 40.38 | 62.11 | 52.50 |
| Titans (MAL) | 19.07 | 20.33 | 40.05 | 69.99 | 48.85 | 53.02 | 67.61 | 35.65 | 40.57 | 61.72 | 52.29 |
| | | | | 1.3B params / 100B tokens | | | | | | | |
| Transformer++ | 18.53 | 18.32 | 42.60 | 70.02 | 50.23 | 53.51 | 68.83 | 35.10 | 40.66 | 57.09 | 52.25 |
| RetNet | 19.08 | 17.27 | 40.52 | 70.07 | 49.16 | 54.14 | 67.34 | 33.78 | 40.78 | 60.39 | 52.02 |
| Mamba2 | 16.56 | 12.56 | 45.66 | 71.87 | 55.67 | 55.24 | 72.47 | 37.88 | 40.20 | 60.13 | 54.89 |
| DeltaNet | 17.71 | 16.88 | 42.46 | 70.72 | 50.93 | 53.35 | 68.47 | 35.66 | 40.22 | 55.29 | 52.14 |
| Gated DeltaNet | 16.42 | 12.17 | 46.65 | 72.25 | 55.76 | 57.45 | 71.21 | 38.39 | 40.63 | 60.24 | 55.32 |
| Samba* | 16.13 | 13.29 | 44.94 | 70.94 | 53.42 | 55.56 | 68.81 | 36.17 | 39.96 | 62.11 | 54.00 |
| Gated DeltaNet-H2* | 15.91 | 12.55 | 48.76 | 72.19 | 56.88 | 57.77 | 71.33 | 39.07 | 41.91 | 61.55 | 56.18 |
| Titans (LMM) | 15.60 | 11.41 | 49.14 | 73.09 | 56.31 | 59.81 | 72.43 | 40.82 | 42.05 | 60.97 | 56.82 |

Table 2: Performance of Titans and baselines on S-NIAH task from RULER benchmark. The best results among simple and hybrid models are highlighted.

| Model | S-NIAH-PK | | | | S-NIAH-N | | | | S-NIAH-W | | |
|---|---|---|---|---|---|---|---|---|---|---|---|
| | 2K | 4K | 8K | 16K | 2K | 4K | 8K | 16K | 2K | 4K | 8K |
| TTT | 98.4 | 98.8 | 98.0 | 88.4 | 60.2 | 36.6 | 10.2 | 4.4 | 78.8 | 28.0 | 4.4 |
| Mamba2 | 98.6 | 61.4 | 31.0 | 5.4 | 98.4 | 55.8 | 14.2 | 0.0 | 42.2 | 4.2 | 0.0 |
| DeltaNet | 96.8 | 98.8 | 98.6 | 71.4 | 47.2 | 15.4 | 12.8 | 5.4 | 46.2 | 20.0 | 1.6 |
| Titans (LMM) | 99.8 | 98.4 | 98.2 | 96.2 | 100.0 | 99.8 | 93.4 | 80.2 | 90.4 | 89.4 | 85.8 |
| Samba | 98.8 | 98.0 | 97.4 | 97.2 | 98.8 | 98.6 | 96.2 | 95.6 | 96.8 | 90.0 | 84.0 |
| Gated DeltaNet-H2* | 99.2 | 97.8 | 97.4 | 98.4 | 98.0 | 97.8 | 96.2 | 95.8 | 97.4 | 96.8 | 88.4 |
| Titans (MAC) | 99.2 | 98.8 | 99.0 | 98.4 | 99.6 | 98.2 | 97.6 | 97.4 | 98.2 | 98.2 | 95.6 |
| Titans (MAG) | 99.4 | 98.0 | 97.4 | 97.4 | 99.2 | 98.8 | 97.2 | 98.6 | 98.0 | 98.0 | 90.2 |
| Titans (MAL) | 98.8 | 98.6 | 98.8 | 97.8 | 99.8 | 98.1 | 96.8 | 96.4 | 98.0 | 97.4 | 92.0 |

memory module achieves the best performance in both perplexity and accuracy measures. Comparing our neural memory module and TTT, which is also a gradient-based recurrent model can show us the importance of our weight decay as well as the momentum. As discussed earlier, the weight decay can be interpreted as a gating mechanism to forget the past data, when it is needed. Also, momentum can help us better manage the memory by providing additional memory for the surprise metric. While some baselines also take advantage of gating mechanism, e.g., Mamba, Mamba2, and Gated DeltaNet, the superior performance of our neural memory module shows the importance of both our surprise mechanism and having deep and non-linear memory. We further discuss the later in Section 4.4.

Comparing the hybrid models, we found that all three variants of Titans (MAC, MAG, and MAL) outperform both Samba (Mamba + attention) and Gated DeltaNet-H2 (Gated DeltaNet + atttention). We attribute the superior performance of Titans (MAL) to the power of neural memory module as the architecture design and used attention are all the same. Comparing Titans (MAG) and (MAC), we find that while their performance are close, MAC performs better when dealing with longer dependencies in the data. Interestingly, both MAG and MAC outperform MAL variant, which due to using the same modules, we attribute this to the architecture design of these models. This finding is particularly important as the current hybrid models (except Hymba [40]) in the literature are using MAL-style combination of recurrent models and attention.

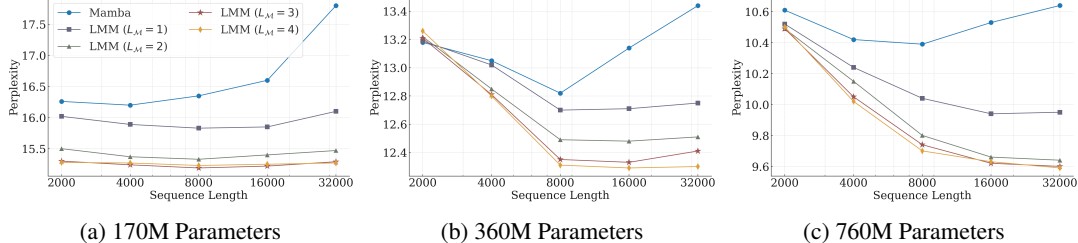

| (a) 170M Parameters | (b) 360M Parameters | (c) 760M Parameters |

Figure 3: The effect of memory depth on the perplexity. Deeper long-term memory results in better scaling in longer sequences.

Table 3: The results of RULER benchmark on hybrid or Transformer-based models.

|  | 16K | 32K | 64K | 128K | Avg. |
|---|---|---|---|---|---|
| Transformer | 81.4 | 79.2 | 63.4 | 49.8 | 68.5 |
| Samba | 83.2 | 80.8 | 69.4 | 34.6 | 67.0 |
| LongRoPE (Phi3-3.8B) | 90.8 | 87.7 | 79.8 | 65.3 | 80.9 |
| EM-LLM | 89.4 | 86.2 | 80.4 | 76.8 | 83.2 |
| Titans (MAC) | 92.0 | 89.2 | 85.2 | 80.4 | 86.7 |

## 4.3 Needle in a Haystack

Scaling a model to longer context window is not always equivalent to being effective for very long sequences [62]. The needle-in-a-haystack (NIAH) task is designed to measure the actual effective context length of models. In this task, we evaluate the model on retrieving a piece of information (i.e., the "needle") from long distractor texts (i.e., the "haystack"). In this part, we use Single NIAH (S-NIAH) task from RULER benchmark [62] and evaluate Titans and baselines on sequences with length 2K, 4K, 8K, and 16K. The results are reported in Table 2. Neural Memory module achieves the best results compare to baselines in all three tasks. We attribute this superior performance to three key differences of Titans with existing sequence models: (1) Compared to TTT, our Neural Memory can better handle the memory capacity by using momentum and also the forgetting mechanism (i.e., weight decay). Therefore, with increasing the sequence length, the performance of Neural Memory does not drop and show a consistent trend; (2) Compared to Mamba2, which has the gating (forgetting) mechanism, Titans have deep non-linear memory, resulting in better memory management. Also, contrary to our neural memory and DeltaNet, Mamba2 is not capable of removing a memory and so we can see a significant drop in performance when increasing the sequence length; (3) Compared to DeltaNet, although it is capable of removing memory using delta rule, it cannot erase the memory, lacking forgetting mechanism. Finally, As expected we can see on par or better results when using Titans variants, where the best results correspond to MAC.

Next, we compare the larger and hybrid MAC variant of Titans with other hybrid or long-context enhance models on harder setups of RULER benchmark. The results are reported in Table 3.

## 4.4 The Effect of Deep Memory

In this section, we evaluate the effect of deep memory in both wall-clock training time and model performance[1]. To this end, we focus on different variants of our neural memory module, where $L_{\mathcal{M}} = 1, 2, 3, 4$. We also use Mamba as a baseline for the model performance. For a fair comparison, we use the same training process for all models and train them on a subset of the Pile dataset [63].

We report the perplexity of our models and baselines as the function of the sequence length in Figure 3. Interestingly, with the increase of memory depth, $L_{\mathcal{M}}$, the model can achieve better perplexity over all sequence length. Also, deeper memory modules are more robust to the sequence length when the model has less number of parameters. With the increase of the number of parameters, all models show better performance on longer sequences.

---

[1]Note that, in this experiment, we only focus on the neural memory module to evaluate the effect of memory depth in the memorization process. Combining neural memory with attention as we do in Titans variants, can additionally enhance the performance of the model over long sequences.

Table 4: Ablation Study on Titans. All components of Titans are positively contributing to its performance.

| Model | Language Modeling ppl ↓ | Reasoning acc ↑ | Long Context acc ↑ |
|---|---|---|---|
| LMM | 27.01 | 47.83 | 92.68 |
| +Attn (MAC) | 26.67 | 48.65 | 97.95 |
| +Attn (MAG) | 25.70 | 48.60 | 96.70 |
| +Attn (MAL) | 25.91 | 47.87 | 96.91 |
| Linear Memory | 28.49 | 46.97 | 85.34 |
| w/o Convolution | 28.73 | 45.82 | 90.28 |
| w/o Momentum | 28.98 | 45.49 | 87.12 |
| w/o Weight Decay | 29.04 | 45.11 | 85.60 |
| w/o Persistent Memory | 27.63 | 46.35 | 92.49 |

## 4.5 Ablation Study

Finally, we perform ablation studies on the different architectural choices in Titans. We consider our neural memory module as a base model and then changing one component at a time: (1) replacing deep memory with linear memory, removing (2) convolution, (3) momentum in the surprise measure, (4) weight decay (or forgot mechanism), and (5) persistent memory. The results are reported in Table 4. All components of neural memory design are positively contributing to its performance, where the greatest contribution comes from weight decay, momentum, convolution, and persistent memory, respectively.

**The Effect of Architectural Design.** To evaluate the effect of architecture design, we compare the performance of three represented variants of Titans in three aspects of (i) language modeling, (ii) commen-sense reasoning, and (iii) long context NIAH (BABILong) tasks. The results are reported in Table 4. MAC and MAG have close performance in language modeling and common-sense reasoning tasks, while MAC achieve significantly better performance in long-context NIAH. Both achieve better performance than MAL. These results along with Figure 10, show a trade-off between fast training and more expressive design.

We provide additional experimental results, including efficiency evaluation, BABILong benchmark, the effect of deep memory as well as Titans performance on state tracking tasks in Appendix G.

## 5 Conclusion

In this paper, we present a neural long-term memory that, as a meta in-context learner, learns to memorize at test time. The neural memory module is a recurrent model in nature, and is adaptively memorizing tokens that are more surprising or are close to surprising tokens. Using this memory, we present Titans architectures, and its three variants, in which we suggest to incorporate the memory module as (1) a context, (2) gating, and (3) a layer. Our experimental evaluation on diverse tasks tasks validate that Titans are more effective than Transformers and recent modern linear recurrent models, specifically for long context.

**Limitations.** Next, we highlight some limitations and potential future directions:

- First, the long-term neural memory is a test time memorizer and so the in-context data is compressed into its parameters. In some cases, this memorization can lead to security, safety, and alignment challenges. Future study can be to understand if the long-term memory can cause security, safety, and alignment issues, and if so, how one can mitigate it.

- The main focus of our paper has been on the design of long-term memory system that can enhance the performance of the attention as a short-term memory. There are, however, several important aspects that requires further study. For example, we leave the theoretical guarantee on the capacity and memory management of Titans as a future study. It is notable that after the initial public version of this work, Behrouz et al. [64] have provided theoretical results on the memory capacity and memory management of deep memory modules.

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

# A  Preliminaries

In this section, we discuss the notation and some background concepts that we use through the paper. We let $x \in \mathbb{R}^{N \times d_{\text{in}}}$ be the input, $\mathcal{M}$ be a neural network (neural memory module), $\mathbf{Q}, \mathbf{K}, \mathbf{V}$ be the query, key and value of the attention mechanism, and $\mathbf{M}$ be the attention mask. When segmenting the sequence, we use $\mathbf{S}^{(i)}$ to refer to the $i$-th segment. Through the paper, we abuse the notation and use subscripts to refer to a specific element of a matrix, vector, or segments. For example, we let $\mathbf{S}_j^{(i)}$ be the $j$-th token in the $i$-th segment. The only exception is subscripts with $t$, which we reserved to index recurrence over time, or the state of a neural network at time $t$. Given a neural network $\mathcal{N}$ and a data sample $x$, we use $\mathcal{N}(x)$ (resp. $\mathcal{N}^*(x)$) to refer to the forward pass with (resp. without) weight adjustment. Also, we abuse the notation and use $\mathcal{N}^{(k)}$ to refer to the $k$-th layer of the neural network. In the following, we first, discuss the backgrounds for attention and its efficient variants followed by a review of modern linear RNNs. Finally, we discuss a memory perspective of these architectures that motivates us to design Titans.

## A.1  Backgrounds

**Attention.** Transformers [1] as the de facto backbone for many deep learning models are based on attention mechanism. Given input $x \in \mathbb{R}^{N \times d_{\text{in}}}$, causal attention computes output $\mathbf{y} \in \mathbb{R}^{N \times d_{\text{in}}}$ based on softmax over input dependent key, value, and query matrices:

$$\mathbf{Q} = x\mathbf{W_Q}, \qquad \mathbf{K} = x\mathbf{W_K}, \qquad \mathbf{V} = x\mathbf{W_V}, \tag{14}$$

$$\mathbf{y}_i = \sum_{j=1}^{i} \frac{\exp\left(\mathbf{Q}_i^\top \mathbf{K}_j / \sqrt{d_{\text{in}}}\right) \mathbf{V}_j}{\sum_{\ell=1}^{i} \exp\left(\mathbf{Q}_i^\top \mathbf{K}_\ell / \sqrt{d_{\text{in}}}\right)}, \tag{15}$$

where $\mathbf{W_Q}, \mathbf{W_K}$, and $\mathbf{W_V} \in \mathbb{R}^{d_{\text{in}} \times d_{\text{in}}}$ are learnable parameters. Despite the power and effectiveness in recall, transformers need at least $N \times d$ operators to calculate the output, resulting in larger memory consumption and lower-throughput for longer sequences.

**Efficient Attentions.** To improve the memory consumption and throughput of softmax attention for longer sequences, various studies focused on I/O aware implementations of attention [65, 66], designing more efficient attention mechanisms by sparsifying the attention matrix [67–69], approximating the softmax [70], or developing kernel-based (linear) attentions [8, 71, 9, 72]. In this part, we focus on the later, i.e., linear attentions, where the softmax in standard attention is replaced with an alternative kernel function $\phi(.,.)$, such that $\phi(x, y) = \phi(x)\phi(y)$. Accordingly, the attention can be written as:

$$\mathbf{y}_i = \sum_{j=1}^{i} \frac{\phi(Q_i^\top K_j)}{\sum_{\ell=1}^{i} \phi(Q_i^\top K_\ell)} V_j = \sum_{j=1}^{i} \frac{\phi(Q_i)^\top \phi(K_j)}{\sum_{\ell=1}^{i} \phi(Q_i)^\top \phi(K_\ell)} V_j = \frac{\phi(Q_i)^\top \sum_{j=1}^{i} \phi(K_j)V_j}{\phi(Q_i)^\top \sum_{\ell=1}^{i} \phi(K_\ell)}, \tag{16}$$

resulting in a higher-throughput as terms $\sum_{j=1}^{i} \phi(K_j)$ and $\sum_{\ell=1}^{i} \phi(K_\ell)$ are re-using in each step. When choosing the kernel as identity matrix [49], the above formulation can also be written in a recurrent format:

$$\mathcal{M}_t = \mathcal{M}_{t-1} + K_t^\top V_t, \tag{17}$$

$$\mathbf{y}_t = Q_t \mathcal{M}_t, \tag{18}$$

which allows efficient inference for linear attentions.

**Modern Linear Models and Their Memory Perspective.** As discussed earlier, one can define learning as a process for acquiring effective and useful memory. Building upon this, one can see the hidden state of Recurrent Neural Networks (RNNs) as a memory unit, which the model aims to compress the information into. Accordingly, in a general form of recurrent neural network, the hidden state can be treated as a memory unit and the recurrence process can be split into the *read* and *write* operations in the memory unit. That is, we let $x \in \mathbb{R}^{N \times d_{\text{in}}}$ be the input, $\mathcal{M} \in \mathbb{R}^d$ is the memory unit, and $\mathbf{y} \in \mathbb{R}^{d_{\text{in}}}$ is the output, then the general form of the recurrent neural network is defined as:

$$\mathcal{M}_t = f(\mathcal{M}_{t-1}, x_t), \qquad\qquad \text{Write Operation} \tag{19}$$

$$\mathbf{y}_t = g(\mathcal{M}_t, x_t), \qquad\qquad \text{Read Operation} \tag{20}$$

where $f(.,.)$ is the *read* and $g(.,.)$ is the *write* corresponding functions. Note that here the subscript of $\mathcal{M}_t$ shows the state of the memory at time $t$.

In this perspective, the recurrence formula of linear Transformers (see Equation 17) is equivalent to additively compress and write keys and values, $(K_t, V_t)$, into a matrix-valued memory unit $\mathcal{M}_t$. Therefore, when dealing with long context data, this additive nature of the process results in memory overflow, significantly damaging the performance of the model. To address this, studies have focused on two promising directions: (1) Adding forget mechanism: several studies have presented adaptive (data-dependent) forgetting gate mechanisms for linear models, where it can erase the memory when it is needed. As examples of such models, we refer to GLA [9], LRU [31], Griffin [73], xLSTM [74], and Mamba2 [30], which the later is also connected to the discretized version of traditional state space models [45].(2) Improving the write operation: To overcome the additive nature of memory write operation in traditional recurrent models, Widrow and Hoff [75] presented Delta Rule, in which before adding a memory (i.e., a pair of key and value), the model first removes its past value. To enhance the parallelizable training and scaling, Yang et al. [38] present a fast paralellizable algorithm. Finally, very recently, Yang et al. [35] improved the DeltaNets by adding a forget gate.

**Memory Modules.** Memory has always been one of the core parts of the neural network designs [76, 12, 77, 32]. The idea of seeing linear layers as the key-value (associative) memory system backs to fast weight programs, in which dynamic fast programs are incorporated into recurrent neural networks to serve as writable memory [76]. The two learning rules of Hebbian [78] and delta [79] are the most popular learning rules for fast weight programs, which have been extensively explored in various studies [80, 76, 81, 71, 82, 38, 35]. All these models, however, are based on momentary surprise, missing the token flow in the sequences (see Section 2.1), and most of them lacks a forgetting gate, resulting in a poor memory management.

We further discuss the connection of our architectures with recent models in Appendix F. Additional related work are discussed in Appendix B.

# B   Related Work

There are diverse perspectives that can independently lead to the design of Titans or its components. Accordingly, to further situate our work in a broader context, we review three categories of studies:

## B.1   Linear Recurrent Models

Recently, to address the computational cost of Transformers in both training and inference, linear recurrent models have attracted much attention [83], mainly due to their fast inference and training. The first generation of models–such as RetNet [49], LRU [31], RWKV [84], S5 [37], and S4 [85]–uses data-independent transition matrix/decay mechanism. The second generation of such models started to incorporate gating mechanism, a widely used techniques in traditional RNNs [86–88], into such linear architectures–e.g., Griffin [73], SSMs [89, 54, 30, 45], RWKV6 [90]. The third generation of linear recurrent models are based on more complex memory updating rule based on meta-learning, online learning, and/or delta-rule, resulting in more expressive and effective models such as: Longhorn [36], Gated DeltaNet [35], TTT [21], and DeltaNet [71]. Our LMM model can be seen as the next generation of such models, in which we incorporate the token flow into the memory updating mechanism, having more powerful memory updating process. See Appendix F for a detailed discussion of different recurrent models and Titans.

## B.2   Transformer-based Architectures

**Transformers.** Transformers [1] as the de facto backbone for many deep learning models are based on attention mechanism [91]. They, however, suffer from quadratic computational cost, limiting their ability to scale to long context window. To improve the memory consumption and throughput of softmax attention for longer sequences, various studies focused on I/O aware implementations of attention [65, 66], designing more efficient attention mechanisms by sparsifying the attention matrix [67–69, 92, 69, 93], approximating the softmax [70], or developing kernel-based (linear) attentions [8, 71, 9, 72].

**Segment-based Transformers.** Another line of research to improve the efficiency of Transformers is segment-based or Chunk Transformers [68]. The main drawback of chunk Transformers is that segments are fully separated and so the context window is limited to the length of the chunks. To address this issue, various studies discuss the importance of a memory so it can help the model to transfer information across chunks [94–99, 98, 100, 101, 96, 97]. The key differences of Titans with these models are: (1) The memory in such models are simple small size vectors, lacking expressive power to compress complex information; (2) The memory module lacks forget mechanism, leading to a fast memory overflow; (3) only focus on momentary surprise, missing the information flow. More specifically, recalling Recurrent Memory Transformers (RMT) [94, 95, 100], one can treat Titans (MAC) as the generalization of RMT, where we use a neural memory module instead of a vector-valued small size memory. In addition to models with fixed-size vector-valued memory, MELODI [102] also discusses more expressive memory system where the memory is updated through append operation over time. This model can be seen as a special case of Titans, where memory is static.

**Memory for Large Language Models.** Another interesting research direction has been to incorporate external memory modules to LLMs after training [103–105]. Such models are different from our approach as we incorporate the memory as a part of initial architecture and so we train it in an end-to-end manner. Also, most of these explicit memory modules suffer from the same limitations as chunk-based Transformers (mentioned above). For a detailed discussion of such models, we refer to the recent study of Wang et al. [106]. Several studies also discuss non-parametric memory modules for Transformers [107, 108]. While these methods are fundamentally different from ours (and potentially can be used as complementary components), we have provided experimental results to compare such methods with Titans in long context tasks. Recently, Zancato et al. [109] also present a hybrid model that uses state-space-models as the long-term memory for attention module. This design is fundamentally different from ours as: (1) Memory: the fading memory is a Mamba, which uses a vector-valued memory, but Titans use a neural deep architecture as the memory, resulting in a potentially more powerful memory. (2) Token Selection: While we use a surprise metric based on gradient descent with momentum (measures the encoding of tokens), B'MOJO uses a dictionary and finds the shortest sequence that is currently unknown to assign. This selection is based on the actual sequence not the encoding (conceptual representation), and so might face issue in memorizing different words with the same concept. (3) In Titans, Memory helps attention to have access to relevant long past, and then attention helps the memory to filter important information of current context (inter-connected memory system), while in B'MOJO, only fading memory is helping attention. (4) Forget gate, (5) Past surprise metric in Titans can consider the flow in the tokens.

## B.3 Test Time Training and Fast Weight Programs

**Memory Design and Augmentation with Memory.** In the literature, a substantial research effort have been toward designing memory modules that are capable of either memorizing the knowledge abstraction (e.g., persistent memory) [39], or memorizing the data-dependent information (also known as contextual memory), through recurrence [97, 94, 95], Transformers [110, 32–34, 111, 99], gradient [81, 23], or other learning paradigms [112, 113]. These memory models, however, either (1) are based on momentary surprise, missing the data flow and events, (2) lack forget mechanisms to remove the memory, leading to a fast memory overflow (3) are fixed-size shallow (matrix valued) memory, resulting in poor performance in long context, and (4) are based on fixed parameters at test time, lacking test time adaption.

**Fast Weight Programs.** The idea of seeing linear layers as the key-value (associative) memory system backs to fast weight programs, in which dynamic fast programs are incorporated into recurrent neural networks to serve as writable memory [71, 76, 114]. The two learning rules of Hebbian [78] and delta [79] are the most popular learning rules for fast weight programs, which have been extensively explored in various studies [80, 76, 81, 71, 82, 38, 35]. All these models, however, are based on momentary surprise, missing the token flow in the sequences (see Section 2.1), and most of them lacks a forgetting gate, resulting in a poor memory management. In this context, deep memory has also been discussed by Irie et al. [82], where they suggest a 2-layer MLP as the memory. However, there are critical differences with Titans: (1) The update rule in Titans is based on both momentary and past surprise, (2) The deep memory in Titans uses input-dependent per-layer parameters, which allows

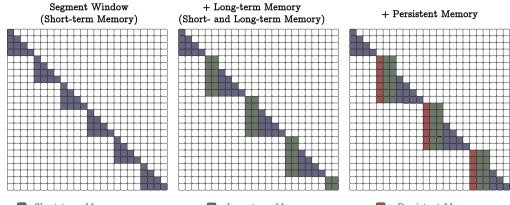
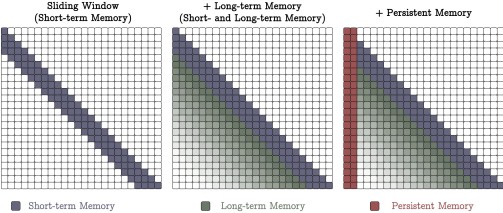

(a) **Memory as a Context (MAC).** We segment the sequence and use full causal attention in each window. Again, the first $N_p$ tokens are persistent memory and the next $N_l$ are long-term memory tokens

(b) **Memory as Gating (MAG).** We use sliding window attention (SWA) as a short-term memory and our neural memory module as a long-term memory, combining by a gating.

Figure 4: Attention masks for different variants of Titans.

each layer flexibly update its weights. Without this, the 2-layer memory do not show improvement in larger scale; (3) the desing of Titans is parallelizable and can be trained in larger scales.

**Test Time Training.** The key ideas of learning at test time or learning to learn (i.e., [115]) backs to very early studies on local learning [116], in which each test data sample is trained on its neighbors before making a prediction [117, 22]. This approach further has shown promising performance in vision tasks [118, 119], mostly due to their ability to mitigate out-of-distribution samples. The most similar studies to ours in this direction are MNM [81] and TTT-layer [21], which we discussed the key differences in Appendix F.

## C  Persistent Memory

Next, we discuss the motivation of persistent memory from three perspective:

**Memory Perspective.** As discussed earlier, our neural long-term memory is a contextual memory, in which all parameters are input-dependent. An effective memory system, however, also needs input-independent parameters to store the abstraction of the task knowledge. That is, mastering a task requires the memorization of the knowledge that how the task can be done, and these parameters are responsible for storing such knowledge.

**Feedforward Network Perspective.** In the Transformer architectures, there are fully connected layers after the attention module, which are shown to be similar to attention weights but with data-independent parameters. That is, Sukhbaatar et al. [39] showed that replacing the `ReLU` in fully connected layers with `Softmax` can results in an attention-like weights, in which weights are data-independent:

$$FFN(x) = W_V \, \texttt{Softmax} \left( W_K x \right). \tag{21}$$

In fact, $W_K$ and $W_V$ are acting similar to $K$ and $V$ matrices in attention module when they are input-independent. The persistent memory weights are expected to have the same functionality, meaning that using them in the first part of the sequence leads to having input-independent attention weights [39].

**Technical Perspective.** Attention with causal mask has implicit bias toward initial tokens in the sequence, and so attention weights are almost always highly active for initial tokens, resulting in performance damage. From the technical perspective, these learnable parameters at the start of the sequence can mitigate such effect by redistributing the attention weights more effectively [120, 121].

## D  How to Incorporate Memory

### D.1  Memory as a Gate

The overall attention mask of this design is shown in Figure 4b. In this design, sliding window attention acts as a precise short-term memory, while the neural memory module is acting as a fading

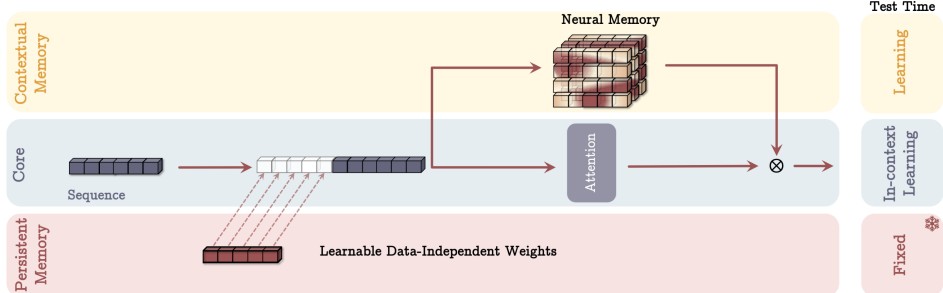

Figure 5: **Memory as a Gate (MAG) Architecture.** This architecture, similarly, has the three branches of (1) core, (2) contextual memory, and (3) persistent memory. It, however, incorporates only persistent memory into the context and combine memory with the core branch using a gating mechanism. At test time, the behavior is the same as Figure 2.

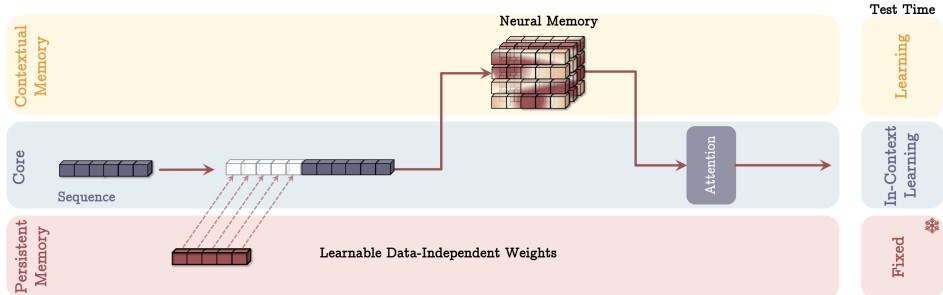

Figure 6: **Memory as a Layer (MAL) Architecture.** In this architecture, the memory layer is responsible to compress the past and current context before the attention module.

memory for the model. This architecture design can also be seen as a multi-head architecture where the structure of heads are different [40].

### D.2 Memory as a Layer

The last variant uses the neural Memory As a Layer (MAL) of a deep neural network (see Figure 6). This architecture design is more common in the literature, where the hybrid models stack recurrent models with full or sliding window attentions. Given input $x$, we have:

$$\tilde{x} = \begin{bmatrix} p_1 & p_2 & \dots & p_{N_p} \end{bmatrix} \;\|\; x, \tag{22}$$

$$y = \mathcal{M}(\tilde{x}), \tag{23}$$

$$o = \texttt{SW-Attn}\left(y\right), \tag{24}$$

where `SW-Attn` is sliding window attention. The main drawback of this design is that the power of the model is limited by each of the layers and so it cannot take advantage of the complementary data processing of attention and neural memory module. In our experiments, for evaluating memory in this design, we use a similar architecture as H3 [122], where we replace the the sequence model with our neural memory module (LMM).

**Memory Without Attention.** Although in the above, we discussed MAL as the combination of LMMs and attention in a sequential manner, one simple variant of MAL is to treat LMM as a sequence model without any attention. From the memory perspective, as discussed in Section 1, we expect each part of the memory system to work independently, even if other components are disturbed. Therefore, a long-term memory module should still be a powerful model even without short-term memory (i.e., attention). We refer to this variant as LMM or Titans (LMM) in our experiments. We provide additional discussions on the connection of Titans and other modern recurrent models in Appendix F.

Table 5: Architectural Details.

| Model | Block | Dim | Head | Peak LR | Token |
|-------|-------|------|------|---------|-------|
| 170M  | 12    | 768  | 16   | 3e-3    | 15B   |
| 340M  | 24    | 1024 | 16   | 1.5e-3  | 15B   |
| 760M  | 24    | 1536 | 16   | 1.25e-3 | 30B   |
| 1.3B  | 18    | 2048 | 8    | 7e-4    | 100B  |

# E  Experimental Details

## E.1  Architecture Details

We report the number of blocks, heads, size of hidden dimension, and peak of learning rate in Table 5. In the training, we follow the training procedure of Yang et al. [35], and use LLama 2 tokenizer with a vocabulary size of 32K and use training length of 4K tokens (2K for SWA). We fixed the persistent memory size (# tokens) to 128, and use 256 memory tokens to encode the past data (i.e., output of the long-term memory). We employ AdamW optimizer with learning rate of $4e\text{-}4$ with cosine annealing schedule with batch size of 0.5M tokens, and weight decay of $0.1$. We use: (1) chunk size: which is 16. The smaller the chunk is the better performance we can get but with the cost of slower model; (2) segment size: in which, we follow previous studies and use 2048 as sliding window (when exists) and 512 as segment size.

## E.2  Language Modeling and Common-sense Reasoning Datasets

Following recent studies on linear recurrent models [35, 30, 38], we use Wikitext [123], LMB [124], PIQA [125], HellaSwag [126], WinoGrande [127], ARC-easy (ARC-e) and ARC-challenge (ARC-c) [128], SIQA [129], and BoolQ [130]. Also, the baselines results for 400M models are from the reported results by Yang et al. [35].

# F  Long-term Memory Module (LMM) as a Sequence Model

In this section, we discuss how LMM as a sequence model is connected to modern linear recurrent models. For the sake of simplicity, we start with a linear memory, where $\mathcal{M}_t = W_t \in \mathbb{R}^{d_{in} \times d_{in}}$. In this case, our objective function becomes $\ell(\mathcal{M}; x_t) = \frac{1}{2} \|\mathcal{M}_t \mathbf{k}_t - \mathbf{v}_t\|_2^2$, in which we use gradient descent with momentum and weight decay for the optimization. Accordingly, revisiting the recurrent formula in Equation 3:

$$\mathcal{M}_t = \texttt{diag}\left(1 - \alpha_t\right)\mathcal{M}_t + S_t \tag{25}$$

$$S_t = \texttt{diag}\left(\eta_t\right)S_{t-1} - \texttt{diag}\left(\theta_t\right)\left(\mathcal{M}_{t-1}\mathbf{k}_t^\top \mathbf{k}_t - \mathbf{v}_t^\top \mathbf{k}_t\right). \tag{26}$$

**LMM is Generalized Gated DeltaNet.** As discussed by Yang et al. [35], DeltaNet [38] can alternatively be interpreted as an online learning problem that optimizes the $\mathcal{L} = \frac{1}{2} \|\mathbf{S}_t \mathbf{k}_t - \mathbf{v}_t\|_2^2$, resulting in:

$$\mathbf{S}_{t+1} = \mathbf{S}_t - \theta_t \nabla \mathcal{L} = \mathbf{S}_t \left(\mathbf{I} - \theta_t \mathbf{k}_t \mathbf{k}_t^\top\right) + \theta_t \mathbf{v}_t \mathbf{k}_t^\top. \tag{27}$$

In this formulation, Gated DeltaNet is the same as above but with an additional weight decay term [35]. Comparing Equation 25 and Equation 27, we can see that setting $\eta_t = 0$ results in both formulations to be equivalent. Accordingly, we can say LMM is generalizing the very recent study of Gated DeltaNet [35] from three aspects:

- Momentum-based Rule: The Delta Rule is based on momentary surprise, meaning that the flow of tokens cannot affect the memory update rule. LMM, however, is based on a momentum rule, which consider *both* past and momentary surprise.

- Deep Memory: While Gated DeltaNet is limited to a linear (matrix-valued) memory as it requires finding the closed recurrence form, LMM allows using deep memory module by using a gradient-based formulation, resulting in higher expressive power.

- Non-Linear Recurrence: While DeltaNet and Gated DeltaNet are based on linear recurrence, our LMM is using inter-chunk non-linear recurrence and intra-chunk linear recurrence. This design allows LMM having a higher expressive power.

- Channel-wise Forget Gate: Contrary to Gated DeltaNet, which uses a shared scalar forget gate across all channels, LMM uses channel-wise forget gate, which is more expressive.

Here, we discussed Gated DeltaNet as a sample of recent generation of recurrent models. Similar approaches such as RWKV-7 [131] are also using the same formulation and loss function, and so LMM is generalizing all such models. Note that RWKV-7 is also using channel-wise forget gate and so is similar to LMM in that manner.

**LMM is Generalized Longhorn.** Similar to DeltaNet, Longhorn [36] uses the same loss function but it derives the closed form using implicit online learning:

$$\mathbf{S}_{t+1} = \mathbf{S}_t \left( \mathbf{I} - \delta_t \mathbf{k}_t \mathbf{k}_t^\top \right) + \delta_t \mathbf{v}_t \mathbf{k}_t^\top,$$

(28)

where $\delta_t = \frac{\theta_t}{1 + \theta_t \mathbf{k}_t \mathbf{k}_t^\top}$. It, however, lacks a forgetting gate, resulting in a faster memory overflow. Therefore, in addition two the abovementioned aspects of (1) Momentum-based Rule, (2) Deep Memory, and (3) Non-Linear Recurrence, LMM has the advantage of using an additional (4) Forget Gate, leading to a better memory management.

**LMM is Generalized TTT Layer.** To the best of our knowledge, TTT [21], is the only modern linear recurrent models with a gradient-based updating rule. In addition to different architectural designs and also objective functions, our LMM has three key differences with presented TTT layers [21]:

1. Forgetting Mechanism: TTT layers are updating memory at each time, without having the chance to forget the past data. Accordingly, when fixing the memory size, the model cannot manage the memory for long sequences. A forget mechanism, such as LMM's, allows clearing the memory when very past information is not needed anymore. We show that in a general case, this forget mechanism is equivalent to weight decay and provide a fast method to incorporate it into the parallel training.

2. Momentum-based Update Rule: TTT layers are based on momentary surprise, meaning that the flow of tokens cannot affect the memory update rule. LMM, however, is based on a momentum rule, which consider *both* past and momentary surprise. See Section 2.1 for the motivation of this design.

3. Deep Memory: While TTT-layers allows for deeper memory, the advantages/disadvantages of such deeper memory modules have not been experimentally evaluated. Also, we observed that without our proposed layer-wise parameters, deep memory cannot show its advantages over shallow memory.

To the best of our knowledge, our neural long-term memory module is the first linear recurrent model with momentum-based update rule.

Finally, as a key difference with all the above and other recent linear recurrent studies, note that the hybrid variants of modern linear models–such as Griffin [73], DeltaNet [38], Gated DeltaNet [35], H3 [122], Mamba2 [30], Samba [132], etc.–all are based on sequential layer-wise design. We present Titans to show how effectively one can incorporate such memory modules into an architecture.

# G  Additional Experimental Results

## G.1  BABILong Benchmark

In this section, we use a hard task from BABILong benchmark [53], in which the model needs to reason across facts distributed in extremely long documents. We follow the original experimental setup and training process in the benchmark. There are two settings: (1) Few-shot setting, in which we use large pre-trained models, and (2) fine-tuning setting, where we fine-tune the MAC variant of Titans to compare it with other fine-tuned baselines. The results for few-shot setting are reported in Figure 9a. In this setup, we can see Titans outperform all baselines–i.e., Mamba2.8B [45],

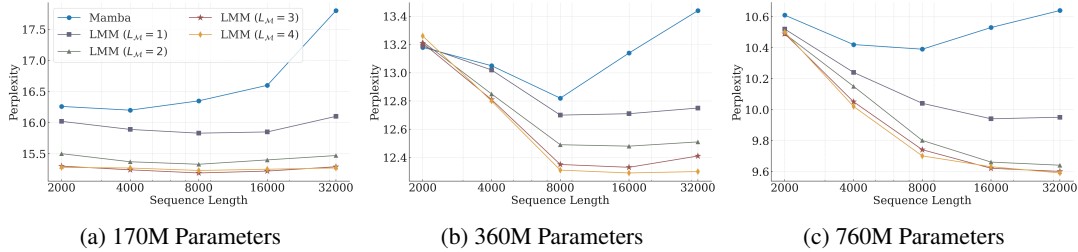

|  |  |  |
|:---:|:---:|:---:|
| (a) 170M Parameters | (b) 360M Parameters | (c) 760M Parameters |

Figure 7: The effect of memory depth on the perplexity. Deeper long-term memory results in better scaling in longer sequences.

RWKV-6-7B [90], RecurrentGemma-9B [51], Gemma-9B [133], Llama3.1-8B [48], GPT-4, and GPT4o-mini [50].

In the fine-tuning setup, we compare the small fine-tuned version of Titans (MAC) with: (i) the fine-tuned version of small models (almost the same number of parameters as Titans) such as Mamba [45], RMT [94], (ii) large models with Retrieval-Augmented Generation (RAG) [134] such as Llama3.1-8B [48], and (iii) extremely large models such as GPT-4 [50], GPT4o-mini, Qwen2.5-72B [135], and Llama3.1-70B [48]. Baseline results are reported by [53]. The results of Titans and baselines are reported in Figure 9b. Titans outperform all models even extremely large models like GPT4. Also, compared to Transformer-based with memory models like RMT, Titans show better performance mainly due to their powerful memory.

It is notable that, in the above we aimed to compare sequence modeling backbones as the choice of different architectures can be orthogonal to this our work. However, there are other choices of architectures that can provide even better performance than Titans in BABILong benchmark. For example, ARMT [95] achieves 98, 98, 98, 98, 98, 98, 97, 95, 93, 77 for 2K, 4K, 8K, 16K, 32K, 64K, 128K, 512K, 1M, 10M sequence length respectively. This performance is better than Titans' results. However, one can simply incorporate our memory module into ARMT update and so achieve a better performance. To validate this claim, we also added a new hybrid variant that uses GPT2 + MAC + MAL. The results are 99, 99, 98, 97, 98,96, 96, 95, 94, 79 for the above sequence length, respectively.

## G.2 Language Modeling and Common-Sense Reasoning

## G.3 The Effect of Deep Memory

In this section, we evaluate the effect of deep memory in both wall-clock training time and model performance[2]. To this end, we focus on different variants of our neural memory module, where $L_{\mathcal{M}} = 1, 2, 3, 4$. We also use Mamba as a baseline for the model performance. For a fair comparison, we use the same training process for all models and train them on a subset of the Pile dataset [63].

We report the perplexity of our models and baselines as the function of the sequence length in Figure 3. Interestingly, with the increase of memory depth, $L_{\mathcal{M}}$, the model can achieve better perplexity over all sequence length. Also, deeper memory modules are more robust to the sequence length when the model has less number of parameters. With the increase of the number of parameters, all models show better performance on longer sequences.

We also evaluate the effect of memory depth ($L_{\mathcal{M}} = 1, 2, 3, 4$) on the training throughput. We report the training throughput (the number of tokens per second) as the function of sequence length in Figure 8. All models scale linearly with respect to the context length (i.e., constant trend in the number of tokens per second with respect to sequence length). Also, by increasing the memory depth, as expected, we can see a linear trend that a deeper memory results in a slower training. Therefore, it is not always efficient to use deeper memory modules, showing a trade-off between effectiveness and efficiency.

---

[2]Note that, in this experiment, we only focus on the neural memory module to evaluate the effect of memory depth in the memorization process. Combining neural memory with attention as we do in Titans variants, can additionally enhance the performance of the model over long sequences.

Table 6: Performance of Titans and recurrent- and Transformer-based baselines on language modeling and common-sense reasoning tasks. Hybrid models are marked with *. The best results among `simple` and `hybrid` models are highlighted.

| Model | Wiki. ppl↓ | LMB. ppl↓ | LMB. acc↑ | PIQA acc↑ | Hella. acc_n↑ | Wino. acc↑ | ARC-e acc↑ | ARC-c acc_n↑ | SIQA acc↑ | BoolQ acc↑ | Avg. ↑ |
|---|---|---|---|---|---|---|---|---|---|---|---|
| **340M params / 15B tokens** | | | | | | | | | | | |
| Transformer++ | 31.52 | 41.08 | 30.76 | 62.98 | 34.76 | 50.53 | 45.21 | 24.05 | 36.81 | 58.24 | 42.92 |
| RetNet | 32.50 | 49.73 | 28.24 | 62.61 | 34.15 | 50.91 | 44.27 | 23.62 | 36.79 | 59.72 | 42.54 |
| GLA | 28.51 | 43.02 | 28.73 | 64.05 | 35.96 | 50.00 | 54.19 | 24.29 | 37.13 | 58.39 | 44.09 |
| Mamba | 30.83 | 40.21 | 29.94 | 63.79 | 35.88 | 49.82 | 49.24 | 24.56 | 35.41 | 60.07 | 43.59 |
| DeltaNet | 28.65 | 47.30 | 28.43 | 63.52 | 35.95 | 49.63 | 52.68 | 25.37 | 37.96 | 58.79 | 44.04 |
| TTT | 27.44 | 34.19 | 30.06 | 63.97 | 35.71 | 50.08 | 53.01 | 26.11 | 37.32 | 59.83 | 44.51 |
| Gated DeltaNet | 27.01 | 30.94 | 34.11 | 63.08 | 38.12 | 51.60 | 55.28 | 26.77 | 34.89 | 59.54 | 45.42 |
| Titans (LMM) | 26.18 | 29.97 | 34.98 | 64.73 | 39.61 | 51.85 | 55.60 | 28.14 | 34.52 | 59.99 | 46.17 |
| Titans (MAC)* | 25.43 | 28.13 | 36.00 | 65.32 | 40.35 | 51.21 | 58.17 | 29.00 | 38.63 | 60.18 | 47.36 |
| Titans (MAG)* | 25.07 | 28.72 | 36.71 | 64.88 | 40.56 | 52.49 | 57.72 | 28.16 | 39.75 | 60.01 | 47.54 |
| Titans (MAL)* | 24.69 | 28.80 | 35.74 | 64.97 | 39.44 | 51.97 | 56.58 | 28.21 | 38.14 | 57.32 | 46.55 |
| **400M params / 15B tokens** | | | | | | | | | | | |
| Transformer++ | 30.63 | 37.37 | 29.64 | 64.27 | 37.72 | 51.53 | 54.95 | 27.36 | 38.07 | 61.59 | 45.64 |
| RetNet | 29.92 | 46.83 | 29.16 | 65.23 | 36.97 | 51.85 | 56.01 | 27.55 | 37.30 | 59.66 | 45.47 |
| HGRN2 | 32.33 | 47.14 | 26.12 | 64.52 | 35.45 | 52.24 | 55.97 | 25.51 | 37.35 | 59.02 | 44.52 |
| GLA | 27.96 | 36.66 | 27.86 | 65.94 | 37.41 | 49.56 | 56.01 | 26.36 | 38.94 | 59.84 | 45.24 |
| Mamba | 29.22 | 39.88 | 29.82 | 65.72 | 37.93 | 50.11 | 58.37 | 26.70 | 37.76 | 61.13 | 45.94 |
| Mamba2 | 26.34 | 33.19 | 32.03 | 65.77 | 39.73 | 52.48 | 59.00 | 27.64 | 37.92 | 60.72 | 46.91 |
| DeltaNet | 27.69 | 44.04 | 29.96 | 64.52 | 37.03 | 50.82 | 56.77 | 27.13 | 38.22 | 60.09 | 45.57 |
| TTT | 26.11 | 31.52 | 33.25 | 65.70 | 39.11 | 51.68 | 58.04 | 28.99 | 38.26 | 59.87 | 46.86 |
| Gated DeltaNet | 25.47 | 29.24 | 34.40 | 65.94 | 40.46 | 51.46 | 59.80 | 28.58 | 37.43 | 60.03 | 47.26 |
| Samba* | 25.32 | 29.47 | 36.86 | 66.09 | 39.24 | 51.45 | 60.12 | 27.20 | 38.68 | 58.22 | 47.23 |
| Gated DeltaNet-H2* | 24.19 | 28.09 | 36.77 | 66.43 | 40.79 | 52.17 | 59.55 | 29.09 | 39.04 | 58.56 | 47.69 |
| Titans (LMM) | 25.03 | 28.99 | 35.21 | 65.85 | 40.91 | 52.19 | 59.97 | 29.20 | 38.74 | 60.85 | 47.83 |
| Titans (MAC)* | 25.61 | 27.73 | 36.92 | 66.39 | 41.18 | 52.80 | 60.24 | 29.69 | 40.07 | 61.93 | 48.65 |
| Titans (MAG)* | 23.59 | 27.81 | 37.24 | 66.80 | 40.92 | 53.21 | 60.01 | 29.45 | 39.91 | 61.28 | 48.60 |
| Titans (MAL)* | 23.93 | 27.89 | 36.84 | 66.29 | 40.74 | 52.26 | 59.85 | 29.71 | 38.92 | 58.40 | 47.87 |
| **760M params / 30B tokens** | | | | | | | | | | | |
| Transformer++ | 25.21 | 27.64 | 35.78 | 66.92 | 42.19 | 51.95 | 60.38 | 32.46 | 39.51 | 60.37 | 48.69 |
| RetNet | 26.08 | 24.45 | 34.51 | 67.19 | 41.63 | 52.09 | 63.17 | 32.78 | 38.36 | 57.92 | 48.46 |
| Mamba | 28.12 | 23.96 | 32.80 | 66.04 | 39.15 | 52.38 | 61.49 | 30.34 | 37.96 | 57.62 | 47.22 |
| Mamba2 | 22.94 | 28.37 | 33.54 | 67.90 | 42.71 | 49.77 | 63.48 | 31.09 | 40.06 | 58.15 | 48.34 |
| DeltaNet | 24.37 | 24.60 | 37.06 | 66.93 | 41.98 | 50.65 | 64.87 | 31.39 | 39.88 | 59.02 | 48.97 |
| TTT | 24.17 | 23.51 | 34.74 | 67.25 | 43.92 | 50.99 | 64.53 | 33.81 | 40.16 | 59.58 | 47.32 |
| Gated DeltaNet | 21.18 | 22.09 | 35.54 | 68.01 | 44.95 | 50.73 | 66.87 | 33.09 | 39.21 | 59.14 | 49.69 |
| Samba* | 20.63 | 22.71 | 39.72 | 69.19 | 47.35 | 52.01 | 66.92 | 33.20 | 38.98 | 61.24 | 51.08 |
| Gated DeltaNet-H2* | 19.88 | 20.83 | 39.18 | 68.95 | 48.22 | 52.57 | 67.01 | 35.49 | 39.39 | 61.11 | 51.49 |
| Titans (LMM) | 20.04 | 21.96 | 37.40 | 69.28 | 48.46 | 52.27 | 66.31 | 35.84 | 40.13 | 62.76 | 51.56 |
| Titans (MAC) | 19.93 | 20.12 | 39.62 | 70.46 | 49.01 | 53.18 | 67.86 | 36.01 | 41.87 | 62.05 | 52.51 |
| Titans (MAG) | 18.61 | 19.86 | 40.98 | 70.25 | 48.94 | 52.89 | 68.23 | 36.19 | 40.38 | 62.11 | 52.50 |
| Titans (MAL) | 19.07 | 20.33 | 40.05 | 69.99 | 48.82 | 53.02 | 67.54 | 35.65 | 30.98 | 61.72 | 50.97 |
| **1.3B params / 100B tokens** | | | | | | | | | | | |
| Transformer++ | 18.53 | 18.32 | 42.60 | 70.02 | 50.23 | 53.51 | 68.83 | 35.10 | 40.66 | 57.09 | 52.25 |
| RetNet | 19.08 | 17.27 | 40.52 | 70.07 | 49.16 | 54.14 | 67.34 | 33.78 | 40.78 | 60.39 | 52.02 |
| Mamba2 | 16.56 | 12.56 | 45.66 | 71.87 | 55.67 | 55.24 | 72.47 | 37.88 | 40.20 | 60.13 | 54.89 |
| DeltaNet | 17.71 | 16.88 | 42.46 | 70.72 | 50.93 | 53.35 | 68.47 | 35.66 | 40.22 | 55.29 | 52.14 |
| Gated DeltaNet | 16.42 | 12.17 | 46.65 | 72.25 | 55.76 | 57.45 | 71.21 | 38.39 | 40.63 | 60.24 | 55.32 |
| Samba* | 16.13 | 13.29 | 44.94 | 70.94 | 53.42 | 55.56 | 68.81 | 36.17 | 39.96 | 62.11 | 54.00 |
| Gated DeltaNet-H2* | 15.91 | 12.55 | 48.76 | 72.19 | 56.88 | 57.77 | 71.33 | 39.07 | 41.91 | 61.55 | 56.18 |
| Titans (LMM) | 15.60 | 11.41 | 49.14 | 73.09 | 56.31 | 59.81 | 72.43 | 40.82 | 42.05 | 60.97 | 56.82 |
| Titans (MAC) | 14.98 | 11.19 | 49.72 | 73.56 | 57.10 | 59.47 | 72.95 | 41.96 | 42.12 | 60.74 | 57.32 |
| Titans (MAL) | 15.13 | 11.28 | 50.10 | 73.29 | 56.74 | 59.52 | 73.09 | 41.37 | 41.88 | 61.09 | 57.14 |

## G.4 Efficiency

In this part, we compare the efficiency of our neural memory as well as Titans with state-of-the-art sequence models. The training throughput of models for different `sequence length` × `batch size` are reported in Figure 10. Comparing recurrent models, including our neural memory module, we can see our memory module is slightly slower than Mamba2 and Gated DeltaNet, mainly due to: (1) having deep memory and more expressive transition process (memory update), and (2) highly optimized kernel in the implementation of Mamba2. Interestingly, Titans (MAL) are faster than baselines as well as the memory module. The main reason for this better throughput is the highly optimized kernel of Flash-Attention [66], which is used for implementing SWA and full attention module in Titans.

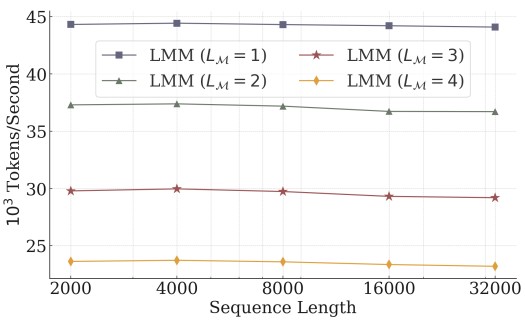

Figure 8: The effect of memory depth on training throughput

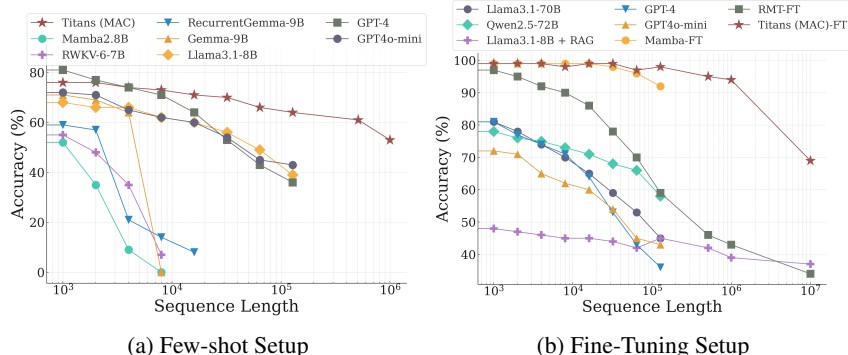

(a) Few-shot Setup  (b) Fine-Tuning Setup

Figure 9: Performance of Titans and baselines on BABILong benchmark. Titans (MAC) outperforms all baselines, including extremely large models, e.g., GPT4. We highlight the fact that ultra-large architectures (i.e., Llama, Qwen, GPT-4) are perform in zero-shot setting, while other small models (i.e., Titans, Mamba, RMT) are fine-tuned. For this, we have followed the original setup in BABILong benchmark [53]

Table 7: Performance on long-term forecasting. The best results are highlighted.

| | Neural Memory | | Simba | | iTransformer | | RLinear | | PatchTST | | Crossformer | | TiDE | | TimesNet | | DLinear | |
|---|---|---|---|---|---|---|---|---|---|---|---|---|---|---|---|---|---|---|
| | MSE | MAE | MSE | MAE | MSE | MAE | MSE | MAE | MSE | MAE | MSE | MAE | MSE | MAE | MSE | MAE | MSE | MAE |
| ETTm1 | 0.358 | 0.387 | 0.383 | 0.396 | 0.407 | 0.410 | 0.414 | 0.407 | 0.387 | 0.400 | 0.513 | 0.496 | 0.419 | 0.419 | 0.400 | 0.406 | 0.403 | 0.407 |
| ETTm2 | 0.261 | 0.309 | 0.271 | 0.327 | 0.288 | 0.332 | 0.286 | 0.327 | 0.281 | 0.326 | 0.757 | 0.610 | 0.358 | 0.404 | 0.291 | 0.333 | 0.350 | 0.401 |
| ETTh1 | 0.420 | 0.421 | 0.441 | 0.432 | 0.454 | 0.447 | 0.446 | 0.434 | 0.469 | 0.454 | 0.529 | 0.522 | 0.541 | 0.507 | 0.458 | 0.450 | 0.456 | 0.452 |
| ETTh2 | 0.336 | 0.382 | 0.361 | 0.391 | 0.383 | 0.407 | 0.374 | 0.398 | 0.387 | 0.407 | 0.942 | 0.684 | 0.611 | 0.550 | 0.414 | 0.427 | 0.559 | 0.515 |
| ECL | 0.162 | 0.261 | 0.169 | 0.274 | 0.178 | 0.270 | 0.219 | 0.298 | 0.205 | 0.290 | 0.244 | 0.334 | 0.251 | 0.344 | 0.192 | 0.295 | 0.212 | 0.300 |
| Traffic | 0.415 | 0.289 | 0.493 | 0.291 | 0.428 | 0.282 | 0.626 | 0.378 | 0.481 | 0.304 | 0.550 | 0.304 | 0.760 | 0.473 | 0.620 | 0.336 | 0.625 | 0.383 |
| Weather | 0.231 | 0.265 | 0.255 | 0.280 | 0.258 | 0.278 | 0.272 | 0.291 | 0.259 | 0.281 | 0.259 | 0.315 | 0.271 | 0.320 | 0.259 | 0.287 | 0.265 | 0.317 |

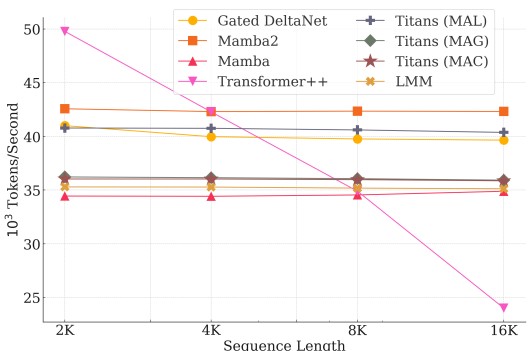

Figure 10: Training throughput comparison of Titans and baselines.

Table 8: Downstream evaluation of pre-trained DNA models on GenomicsBenchmarks [136]. We report top-1 classification accuracy (%).

| Model | Enhancer Cohn | Enhancer Ens | Human Reg. | Non-TATA Promoters | Human OCR Ens. |
|---|---|---|---|---|---|
| CNN | 69.5 | 68.9 | 93.3 | 84.6 | 68.0 |
| DNABERT | 74.0 | 85.7 | 88.1 | 85.6 | 75.1 |
| GPT | 70.5 | 83.5 | 91.5 | 87.7 | 73.0 |
| HyenaDNA | 74.2 | 89.2 | 93.8 | 96.6 | 80.9 |
| Transformer++ | 73.4 | 89.5 | 89.9 | 94.4 | 79.5 |
| Mamba | 73.0 | - | - | 96.6 | - |
| Based | 74.6 | 89.5 | 89.5 | 96.8 | 79.0 |
| Neural Memory Module | 75.2 | 89.6 | 89.3 | 96.6 | 79.9 |

Table 9: The performance of our model compared to baselines.

| | SWDE | NQ | DROP | FDA | SQUAD | TQA |
|---|---|---|---|---|---|---|
| Transformers | 84.9 | 23.0 | 28.4 | 72.5 | 48.1 | 64.4 |
| Gated DeltaNet | 63.2 | 19.1 | 26.7 | 33.4 | 39.6 | 59.7 |
| Titans (LMM) | 65.1 | 20.7 | 27.2 | 37.3 | 42.6 | 61.0 |

## G.5 Time Series Forecasting

To show the effectiveness of our memory module in a broader tasks, we also evaluate its performance in time series forecasting tasks. To this end, we use Simba framework [41] for time series forecasting, and replace its Mamba module with our neural memory. We report the results on common time series forecasting benchmark datasets–ETT, ECL, Traffic, and Weather [6]. The results are reported in Table 7. Our neural memory module is outperforming all baselines, including Mamba-based, linear-based, and Transformer-based architectures.

## G.6 DNA Modeling

In order to understand the capability of Titans beyond natural language, we further evaluate the performance of our neural memory module on DNA modeling tasks. To this end, we evaluate pre-trained models on the downstream tasks in GenomicsBenchmarks [136]. We follow the same experimental setups from Nguyen et al. [137], and re-use the reported results of baselines by Arora et al. [70]. The performance of Titans (LMM) and baselines are reported in Table 8. We find that LMM is competitive with state-of-the-art architectures across different downstream genomics tasks.

## G.7 In-context Recall Tasks

In-context recall is one of the challenging benchmarks for recurrent neural networks. In this section, we show that our Titan due to its design of surprise performs favorably compared to other recurrent neural networks (note that Gated DeltaNet itself outperforms other modern recurrent neural networks), achieving a closer performance to Transformers.

## G.8 Synthetic Benchmark of MAD

We evaluate the performance of Titans (LMM) on MAD benchmark, a synthetic benchmark that evaluate the performance of models in recall, memorization, compression, and copying tasks [138]. The results are reported in Table 10.

Table 10: Performance of Titan and baselines on the synthetic benchmark of MAD [138]. Titan outperforms all the baselines, including Transformers.

|  | Compression | (Noisy) ICR | Fuzzy ICR | Selective Copying | Memorization |
|---|---|---|---|---|---|
| Transformers | 49.4 | 100 | 48.2 | 95.9 | 83.8 |
| Gated DeltaNet | 44.8 | 100 | 32.5 | 96.2 | 81.7 |
| Titans | 49.6 | 100 | 49.7 | 99.4 | 83.5 |

