# OpenReview forum: "Titans: Learning to Memorize at Test Time"
_NeurIPS.cc/2025/Conference — NeurIPS 2025 poster_

### Official Review · Reviewer_WL32 · 2025-06-29

**Clarity:** 2
**Significance:** 3
**Originality:** 2
**Rating:** 5
**Confidence:** 4

**Summary:**

I previously reviewed this paper for ICML 2025, and this review takes into account both the original submission and how the authors addressed reviewer concerns. The paper introduces Titans, a family of architectures incorporating a neural long-term memory module that learns to memorise at test time through gradient-based updates. The key innovation is a surprise-based memory mechanism that combines "momentary surprise" (gradient updates) with "past surprise" (momentum) to determine what information to store. Unlike traditional recurrent models that compress everything into fixed-size states, Titans uses a deep MLP as memory that can adaptively learn and forget information. The authors present three architectural variants that incorporate this memory differently: Memory as Context (MAC) treats memory as additional context for attention, Memory as Gate (MAG) combines memory with sliding-window attention through gating, and Memory as Layer (MAL) uses memory and attention sequentially. The paper demonstrates improvements over baseline models on language modelling, common-sense reasoning, and needle-in-a-haystack tasks, with the authors claiming effectiveness on long contexts.

The authors have made significant efforts to address reviewer feedback, including adding comparisons with hybrid models like Samba, providing more architectural details, expanding the related work section, and adding state-tracking evaluations. However, many critical concerns remain unaddressed as discussed below.

**Questions:**

Since I have many things to say for this paper, I tried to group my most important questions into the following 4 categories about theoretical issues, fair evaluation, positioning and reproducibility:

1. **Theoretical foundations**: Could you provide the theoretical analysis of memory capacity and convergence that was promised in the ICML rebuttal? Additionally, the analysis of 2-3 updates improving performance by 0.5% that you promised to include is missing from the current version - could you add it?
2. **Fair comparative evaluation**: Can you provide (a) BABILong results where all models are either fine-tuned or zero-shot, (b) a unified evaluation where all methods appear on the same benchmarks, and (c) throughput comparisons including Samba in Figure 9? Finally, critical comparisons with key memory baselines are currently relegated to the appendix. To allow for a fair assessment, it is important that competitive results are presented and discussed in the main text. Can you please provide a unified summary table in your rebuttal, and explain how you will restructure the main paper to foreground these essential results in a final version?
3. **Positioning and advantages**: Given that existing surprise-based approaches achieve 100% accuracy at 10M tokens without fine-tuning, what are the specific advantages and use cases for Titans? Why does memory-constrained Titans outperform full Transformers on retrieval tasks?
4. **Reproducibility**: Can you provide clear implementation guidance, including memory size selection criteria, hyperparameter settings, and details about the "highly optimised kernel" mentioned for MAL?

**Ethical Concerns:**

["NO or VERY MINOR ethics concerns only"]

**Final Justification:**

The authors have constructively engaged with the review and committed to addressing my major concerns: adding a theoretical limitations discussion, clarifying the relationship to standard optimisation techniques, moving key long-context results to the main paper, and improving figure labelling for methodological clarity. These changes will significantly strengthen the paper's presentation and contribution. I appreciate the authors' responsiveness and look forward to seeing these improvements in the final version.

**Limitations:**

The authors briefly mention limitations in the appendices but don't adequately address:

- The gap between claimed capabilities ("larger context windows") and actual evaluation (16K tokens)
- Why their approach requires fine-tuning while other surprise-based methods work zero-shot
- Computational overhead of test-time gradient updates
- When their complex architecture is justified over simpler alternatives

The paper would benefit from a dedicated limitations section in the main text.

**Paper Formatting Concerns:**

No formatting concerns.

**Quality:**

3

**Strengths And Weaknesses:**

#### Strengths:
- *Novelty*: Although this work builds on top of a lot of other studies on state-space models as Transformer alternatives, it has some interesting novelties. I found treating these models just as a neural network with gradient updates and momentum very interesting. It is an intuitive connection but no other model I am aware of makes it. Also, the three variants of Titans (MAC, MAG, MAL) provide interesting ways to incorporate memory into sequence models, with MAC showing particularly strong results.
- *Evaluation*: The paper evaluates across multiple domains (language, time series, DNA) showing versatility.
- *Parallelisation*: The chunk-wise training method (Section 2.2) enables practical training despite gradient-based updates, which is crucial for scaling up these models.
- *Performance*: Titans show consistent gains over baselines in multiple benchmarks.
- *ICML feedback*: Authors added Samba comparisons and some other hybrid models in accuracy benchmarks, state-tracking experiments (MAD benchmark), memory interpretability analysis, and expanded related work as requested.

#### Weaknesses:
* Overall quality:
	* Despite promising formal proofs in the ICML rebuttal, the revised paper lacks the theoretical analysis of memory capacity and convergence properties.
	* Some important model comparisons are incomplete. For example, the throughput comparison with Samba is still missing from figures despite specific reviewer requests in ICML.
	* BABILong evaluation remains flawed, comparing fine-tuned small models against zero-shot large models without acknowledging this unfair advantage

* Clarity:
	* "Momentary surprise" is presented as novel when it's identical to the well-established delta rule. I find this misleading as it stands.
	* Different methods are evaluated on different benchmarks without clear justification. Why are some baselines missing from certain benchmarks (e.g. only RULER and not BABILong)? This makes it difficult to form a coherent picture of relative performance.
	* On a related point, in retrieval tasks, methods with no memory constraints (full Transformers, RAG-based or retrieval-based approaches) somehow perform worse than Titans which has explicit memory limitations, despite the fact that in the literature are solving much longer NIAH tasks with 100% accuracy. This surprising result deserves some discussion. Is it due to training differences, evaluation setup, or fundamental advantages of the current approach?
	* While the connection to momentum is mentioned, its framing could be clearer in the main text. The detailed clarification in Appendix F is welcome, but the main narrative would benefit from being more upfront about this connection to a standard optimisation technique
	* There is no clear guidance on hyperparameter selection, memory size choices, or when to use which variant of Titans.

* Significance of claims:
	* Despite claims about "larger context windows," rigorous evaluation (RULER) only extends to 16K tokens.
	* The paper doesn't clearly establish when/why one would use Titans over simpler alternatives or more successful approaches.
	* "Past surprise" is standard momentum; the main contribution appears to be combining existing techniques
	* The paper doesn't adequately distinguish itself from recent work using surprise for memory in LLMs (e.g. EM-LLM).


**Overall,** the paper presents some very interesting architectural ideas and shows empirical improvements. The authors made good-faith efforts to address ICML feedback. However, the presentation remains problematic with key details missing or relegated to appendices without explanation. The revisions give the impression of a checklist being completed rather than a deep re-evaluation of the paper's weaknesses. Critical comparisons lack proper methodology, reproducibility information is scattered, and the theoretical foundations are absent despite being promised. While the core ideas merit publication, the paper needs substantial revision to meet publication standards. My '*Borderline Accept*' score reflects this tension: the core ideas have high potential, but the unaddressed flaws are significant. I believe these issues are addressable, and I look forward to the authors' rebuttal, after which I will re-evaluate my final assessment.

---

> ### Author Rebuttal · Authors · 2025-07-31
>
> **We thank the reviewer for their time and valuable comments**. We respond to your comments below:
>
> > Despite promising formal proofs in the ICML rebuttal, the revised paper lacks the theoretical analysis of memory capacity and convergence properties.
>
> **Response**: We first want to kindly bring to your consideration that providing theoretical results is not a **must** for publications. While we understand the reviewer's interests and curiosity about the theoretical side of memory modules, we want to clarify that none of the peer studies provide such theoretical results about the capacity of the memory or why one-shot memorization works (e.g., GLA (ICML 2023), Mamba2 (ICML 2024), Mamba (COLM 2024, best paper award), Gated DeltaNet (ICLR 2025), GSA (NeurIPS 2024), Simba (NeurIPS 2024), Longhorn (NeurIPS 2024), TTT (ICML 2025, spotlight), etc.). These are indeed interesting questions that one can ask and explore in future studies. Our submission is a 9-page research paper that aims to provide **enough** evidence for its usefulness and effect. Even the initial Transformers paper, after 8 years, is still being studied, improved, and evaluated.
>
> In ICML you mentioned you are willing to increase your initial rating of “weak accept”, if we provide theoretical results. **Despite providing theoretical results and addressing your concerns, your initial rating remained the same**. Therefore, it was not clear for us if our theoretical results are what you expected. In this submission, we decided to make our work more focused, and so leave such exploration for future work, mainly due to the fact that **the goal of this work is not on one-shot memory update nor understanding the capacity of recurrent models**.
>
>
> > Some important model comparisons are incomplete. For example, the throughput comparison with Samba is still missing from figures despite specific reviewer requests in ICML.
>
> **Response**: Please note that as we discussed earlier in the ICML review process, Samba is Attention + Mamba. We already have a throughput comparison with Mamba, which indicates that our Titans (LMM) is faster. Therefore, we can conclude that Samba = Attention + Mamba is slower than our MAL = Attention + LMM. However, following your suggestion, here we provide the Samba throughput and compare it with our models:
>
> | Model | 2K | 4K | 8K | 16K |
> |-| -| -| - |-|
> |Samba|	~39|	~38|	~36|	~35|
> |MAG| ~42|~41|~41|~40|
> |MAL |~43|~43|~41|	~41|
>
>
> We will add these results in the camera ready version of the paper.
>
>
> > BABILong evaluation remains flawed, comparing fine-tuned small models against zero-shot large models without acknowledging this unfair advantage.
>
> **Response**: First, in lines 1374 - 1383 we clearly acknowledged and explained that small models are fine-tuned and large (commercial) models are zero-shot. Therefore, saying ``without acknowledging’’ is a misunderstanding. Second, we have followed the benchmark setups (please see Figure 1 in **Peer-reviewed** BABILong original paper in NeurIPS 2024). Therefore, we believe if there is a setup in this peer-reviewed paper that the reviewer found flawed, we should not be penalized for that. Please note that it is not expected nor we have the budget to re-run all the benchmark results for all the baselines and change the **original** setup of the benchmark. If the reviewer believes that it can solve their concern, we can remove the results of large (commercial) models, which definitely are not expected to compare with.
>
> Moreover, in Table 10, we have provided the results for even 128K sequence length in RULER, which we believe should be enough to show the good performance of Titans in longer contexts.
>
>
> > "Momentary surprise" is presented as novel when it's identical to the well-established delta rule. I find this misleading as it stands.
>
> **Response**: Could you please **specify** the part that we claim momentary surprise as a novel contribution of our work so we can fix it?
>
> In line 1168, we have clearly mentioned that delta-rule is equivalent to momentary surprise and this is the difference of our update rule, because we consider ``past surprise`` as well. Again in line 1328 we clearly mentioned that `The Delta Rule is based on momentary surprise`.
>
>
> > Different methods are evaluated on different benchmarks without clear justification. Why are some baselines missing from certain benchmarks (e.g. only RULER and not BABILong)? This makes it difficult to form a coherent picture of relative performance.
>
> **Response**: Please note that the **only** benchmark with different baselines is the BABILong benchmark. The state-of-the-art recurrent models are considered as the baseline for all language modeling, common-sense reasoning, and RULER tasks. The BABILong benchmark, however, has its own baselines and it has not been considered in any previous similar studies so we can borrow their results. Therefore, it is hard to adapt such models for this downstream task, is out of the scope of our paper, and further requires more compute resources. Once again, please note that we have already shown the good performance of our model until 128K (which has not been evaluated in any similar previous studies on recurrent and linear models). Therefore, if you believe removing BABILong benchmark results can make the evaluation more clear, we would be happy to do it in the camera ready version of the paper.
>
>
> > Despite claims about "larger context windows," rigorous evaluation (RULER) only extends to 16K tokens.
>
> **Response:** This seems to be a misunderstanding. In Table 10, we have already provided the results for the RULER benchmark until 128K sequence length.
>
> Please note that this is a research paper and more than these sequence lengths are often a goal for commercial LLMs, or research papers that are built on large pre-trained models. Furthermore, our goal is to close the gap between the effective context length of recurrent models and they already perform near zero accuracy in 16K context length of RULER.
>
> > "Past surprise" is standard momentum; the main contribution appears to be combining existing techniques
>
> **Response:** We respectfully disagree with the reviewer that this connection affect the novelty of our method. Momentum term has never been used in this context. With the same logic, one can argue that most of recent architecture studies, including the work of DeltaNet (Schlag et al., 2021) and linear attention (Katharopoulos et al., 2020), are not novel because all use gradient descent, which is a well-known algorithm.
>
> > Additionally, the analysis of 2-3 updates improving performance by 0.5% that you promised to include is missing from the current version - could you add it?
>
> **Response:** Yes, we will make sure to add these results in the final version of the paper.
>
> > Fair comparative evaluation: Can you provide (a) BABILong results where all models are either fine-tuned or zero-shot
>
> **Response:** Please note that all small-models are fine-tuned. If the reviewer believes that the original setup of the peer-reviewed benchmark is not proper, we would be happy to remove the results of zero-shot models. Please once again note that all recent studies in this line has focused on RULER, and we have already provided its results until 128K.
>
>
> > Why does memory-constrained Titans outperform full Transformers on retrieval tasks?
>
> **Response:** Please note that, the unbounded memory in Transformers is not always the best for all the tasks. The reason is data can be compressible and so the task does not require direct retrieval. Also, it is notable that unbounded memory in Transformers does not mean that they can utilize it in all tasks. As an example, several papers discuss limitations like ''attention sink’’, which can cause Transformers to not scale to a large context. Also, in Table 8, we have provided the results for retrieval tasks that Transformers achieve better results. Please note that this is one the studies in this direction and requires several follow up works to fully address all the aspects.
>
> Moreover, all Transformer and retrieval-based models consider the dependencies of tokens as linear mapping between keys and values. Even given direct access to all past tokens (having perfect memory) the KV cache directly maps keys to values. This might not always the perfect assumption. On the other hand, Titans uses a deep memory and so can learn non-linear mapping between keys and values.
>
>
> > There is no clear guidance on hyperparameter selection, memory size choices, or when to use which variant of Titans.
>
> **Response:** Please note that memory size is tied to the number of parameters. Given $d$ as the dimension of keys and values, the memory has size $L \times d^2$, where $L$ is the depth of the memory.  Therefore, it is not a new hyperparameter, and can be determined based on the computational resources that are available. Also, in our ablation study, we show that for long-context tasks MAC performs better, while for language modeling MAG achieves the best results. We will emphasize more on these in the final version of the paper. Please let us know if there are specific parts that are not clear.
>
> > Details "highly optimised kernel" mentioned for MAL?
>
> **Response:** Please note that as we mentioned in the paper, this is Flash attention kernel and it is not our contribution. The implementation of attention is based on the highly-optimized kernel of Flash attention.
>
> > Limitations:
>
> **Response:** Following your suggestion, we would be happy to discuss them as the limitations and motivated future work to address them.
>
> > Details of Implementation:
>
> **Response:** Following your suggestion, we adhere to provide all the details upon acceptance of the paper.
>
> **We hope that our above responses have addressed your concerns, and we would be more than happy to answer any remaining concern.**

---

> ### Comment · Program_Chairs · 2025-08-01
> **about re-reviewing**
>
> Dear reviewer,
>
> Re-reviewing is not against policy, provided that the new review reflects the merits of the current version of the paper and does not reveal any names or affiliations. Please adhere to these principles during the discussion and when finalizing your review. Thanks.

---

> ### Author Response · Authors · 2025-08-03
>
> Once again, we want to thank the reviewer for their time and comments. Due to the space constraint, we use this comment to address your questions:
>
> 1. As we discussed above, the theoretical analysis and understanding of the memory capacity is out of the scope of our paper. We understand that it is an interesting future work and will be discuss it in the limitation and future work section. Regarding the performance multiple update of the memory, while we want to bring to your attention that none of the similar studies mentioned above have done that and again it is out of the scope of our paper (because we do not have any claim about multiple update), we would be happy to provide its results in the final version of the paper. We will add this in the final version, following your suggestion.
>
> 2. As we discussed above, all small size models are fine-tuned and only commercial models are zero-shot, which definitely we are not expected to compare with them. However, to fully address your concern, here we provide a comparative evaluation of Titans and baselines we used in other experiments in a fine-tune setup:
>
> | Model | 2K |4K|8K|16K|32K|64K|128K|512K|1M|
> |-|-|-|-|-|-|-|-|-|-|
> Titans| 99| 99|99|97 | 98	|96	|98	|95	|94|
> Mamba| 99|99|99|99 | 98| 97| 93 | 52 | 28 |
> TTT | 99|98|97|97 | 97 | 92 | 88 | 49 | 8 |
> DeltaNet | 99 | 99 | 99 |99 | 96| 92| 90 | 56 | 31 |
>
> Titans show a clear advantage over the baselines.
>
> **Also, we will use the 1 additional page in the camera ready version to bring some results from the appendix to the main part, following your suggestion.**
>
>
> 3. Could you please clarify, which surprise-based model you are referring to? If you mean EM-LLM, please note that we have already compared Titans with EM-LLM in Table 10, until 128K context length and Titans show a clear improvement +3.5% on average (larger context length show better improvement.). Please note that the hardness of each long-context dataset is different.
>
> Regarding the comparison with Transformers, we have clarified that unconstrained memory does not necessarily mean that the model is able to use all that memory effectively. Moreover, some tasks/dataset might not need unconstrained memory and a more powerful memory system could be more useful in such cases. Furthermore, we already have provided additional results in the initial submission on in-context retrieval tasks that show the superior performance of Transformers.
>
> 4. Following your suggestion, we adhere to provide all the details upon acceptance of the paper.
>
>
> ---
> Finally please note that as also acknowledged by you, we have improved the paper from different aspects, modified claims, add additional related work discussions (discussing more than 130 studies!), and have provided additional experiments (while such experiments have not been done by any other study on recurrent models). It is indeed discouraging for us to see you keep your initial rating at ICML, before all these changes. **Please note that there is no study without limitations and there is always room for further evaluation and discussion. However, in a 9-page conference paper, it is expected to provide enough evidence for the usefulness of the approach, which we believe, as acknowledged by you and other reviewers, we have provided.**
>
>
>
> We hope that our above responses have addressed your concerns, and we would be more than happy to answer any remaining concern.

---

> ### Comment · Reviewer_WL32 · 2025-08-04
>
> I'd like to thank the authors for the detailed responses and the additional results provided in your second comment. I appreciate the constructive engagement, particularly the Samba throughput and additional fine-tuned comparisons, and it is encouraging that you agree to move key results from the appendix to the main paper. However, some core concerns remain:
>
> * First, I note that you've chosen to exclude the theoretical analysis citing scope constraints. However, this analysis would significantly strengthen the paper's contributions. The argument that "no other papers provide such theoretical results" is unconvincing - those papers aren't claiming test-time gradient-based memory updates as their core contribution. Since your approach fundamentally relies on gradient descent dynamics for memory formation, some analysis of convergence properties or memory capacity would differentiate this from purely empirical work.
> If such analysis proves intractable, the paper should explicitly discuss why (e.g., "We attempted to analyse convergence properties but found that the interaction between momentum and weight decay in the test-time setting makes formal guarantees difficult to establish because..."). This would be more valuable than omitting the discussion entirely.
>
> * Regarding novelty, thank you for clarifying that momentary surprise equals delta rule (L1168, L1328). However, the paper would benefit from being more upfront about building on standard optimisation techniques. The momentum + delta rule combination is valuable, but shouldn't be obscured by terminology that might suggest more novelty than exists.
>
> * The new fine-tuned results you provide (Titans achieving 94% at 1M vs Mamba's 28%) are particularly compelling and should be prominently featured in the main paper. Also, while you acknowledge the fine-tuned vs zero-shot distinction in lines 1374-1383, this critical methodological detail appears only in the supplementary material. Please consider clearly labelling which models are fine-tuned vs zero-shot directly in all figures.
>
> * Additionally, key results demonstrating your long-context claims remain relegated to appendices (128K RULER results in Table 10, competitive baselines). For a paper whose central contribution involves long-context capabilities, these should be featured prominently in the main text.
>
> For score improvement, I suggest:
> * Adding a limitations paragraph acknowledging theoretical gaps and test-time computational overhead
> * Clarifying in intro (or abstract) that the core innovation is combining existing techniques in a novel architecture
> * Ensuring the planned move of appendix content to the main paper includes your key long-context results (e.g., Table 10).
> * Adding clear fine-tuned vs zero-shot labels directly on all comparison figures or captions that it makes sense.
>
> The paper has merit and will likely be accepted. These changes would significantly strengthen it and address substantive concerns raised across review cycles.
>
> Finally, regarding ICML, I review all submissions based solely on their technical merits. My goal is helping improve the paper through specific, actionable feedback that would benefit readers. Any consistency in technical standards across venues reflects the importance of these particular issues to the work's contribution.

---

> > ### Author Response · Authors · 2025-08-04
> >
> > We thank the reviewer for their time, valuable comments, and engaging with us in the author-reviewer discussion period.
> >
> > Following your suggestion, we adhere to making the following changes in the final version of the paper:
> > 1. We will add a limitation paragraph acknowledging theoretical gaps and also the overhead caused by the test-time update of a deep memory.
> > 2. We clarify in the Introduction that our past surprise metric is equivalent to the momentum term that is commonly used for optimization of neural networks.
> > 3. We will use the additional page in camera ready to also move the long-context results to the main part of the paper.
> > 4. We will add fine-tuned vs zero-shot labels directly to the figures and/or their captions to make sure clarity of the setup.
> >
> > We hope that our above responses has addressed your concerns. We would be happy to clarify any remaining point or discuss any remaining concerns.

---

### Official Review · Reviewer_ajQb · 2025-06-30

**Clarity:** 3
**Significance:** 3
**Originality:** 3
**Rating:** 4
**Confidence:** 4

**Summary:**

The paper introduces Titans, a new family of neural architectures that incorporate a long-term memory module capable of learning to memorize at test time. The core idea is to augment attention mechanisms - which effectively model short-range dependencies but are limited by fixed context lengths - with a trainable neural memory system that resembles human long-term memory. This memory is updated using a "surprise" signal based on gradients during inference and includes a gating mechanism for adaptive forgetting. Three Titans variants are proposed to integrate memory in different ways: as additional context (MAC), through gating (MAG), or as a sequential layer (MAL). These models show improved results across tasks such as language modeling, commonsense reasoning, and long-context benchmarks. The architecture supports parallelizable training and achieves better performance and scalability than several recent recurrent and hybrid Transformer-based models. Titans demonstrate that dynamically updating neural memory at test time can enhance long-range sequence modeling.

**Questions:**

1. Can you clarify how the surprise-based memory update behaves over long sequences? It's unclear how the model avoids drift or forgetting. Some visualization or stability analysis would help.

2. Why use a deep MLP for memory? Have you compared against shallower versions, recurrent cells, or Hopfield-style mechanisms? Ablations would make this design choice more convincing.

3. The paper's relation to test-time training and meta-learning models like TTT or fast weight programmers is underdeveloped. What is the core difference beyond architecture?

4. The role of persistent memory is vague. How is it trained, and how sensitive is the model to its size? More detail would clarify whether it's essential or just another form of prefix tuning.

**Ethical Concerns:**

["NO or VERY MINOR ethics concerns only"]

**Final Justification:**

I found the author's response convincing and have made a small upward adjustment to my evaluation.

**Limitations:**

No. The paper does not adequately discuss limitations or societal impacts. The authors should address possible risks related to test-time adaptation, such as instability, data leakage, or unintended memorization of sensitive information. It would also help to discuss potential computational overhead at inference due to memory updates, and any fairness or privacy concerns when memory is updated dynamically from user data.

**Paper Formatting Concerns:**

-

**Quality:**

3

**Strengths And Weaknesses:**

Quality: The paper presents a technically solid and well-executed approach with a clear motivation for combining attention with a test-time trainable long-term memory. The proposed memory module is grounded in both neuroscience-inspired ideas and practical algorithmic design, and is evaluated across a diverse set of benchmarks. The experiments are thorough, with meaningful ablations and comparisons to relevant baselines. However, some architectural choices, such as the specific structure of the memory MLP or hyperparameter tuning strategies, could have been better justified or explored. While the surprise-based update is interesting, it lacks deeper theoretical analysis or guarantees on memory stability or convergence.

Clarity: The paper is generally well written and structured, but at times suffers from dense exposition and terminology that could challenge readers unfamiliar with meta-learning or memory systems. While figures are helpful, some architectural descriptions are overly abstract or spread across too many sections. The treatment of related work is extensive but might benefit from a clearer contrast with specific prior methods, especially regarding what exactly distinguishes Titans from prior hybrid models. More intuitive explanation of the gating mechanisms or test-time dynamics would improve accessibility.

Significance: The work addresses a central challenge in sequence modeling - how to scale models to long contexts without compromising performance or efficiency. The results demonstrate that Titans outperform strong baselines across tasks, including those focused on long-range reasoning. This suggests potential for practical impact and further adoption. That said, the gains over recent hybrid models like Gated DeltaNet or Samba are sometimes modest, and it remains to be seen how Titans generalize beyond the tasks shown. The design space for combining memory and attention is still broad, and the paper only explores three specific configurations.

Originality: The idea of learning memory at test time via surprise-driven updates is novel in the context of modern sequence models, and the framing of memory as a dynamic, test-time learning module is fresh. While components like gating, persistent memory, and associative loss are not individually new, their combination and positioning within a scalable architecture are meaningfully different. Still, the paper could more explicitly position itself relative to test-time training literature and clarify how it advances those ideas in the long-context modeling setting.

---

> ### Author Rebuttal · Authors · 2025-07-31
>
> **We thank the reviewer for their time and valuable comments**. We respond to your comments below:
>
> > However, some architectural choices, such as the specific structure of the memory MLP or hyperparameter tuning strategies, could have been better justified or explored.
> > Why use a deep MLP for memory? Have you compared against shallower versions, recurrent cells, or Hopfield-style mechanisms? Ablations would make this design choice more convincing.
>
> **Response:** Thank you for mentioning this. However, please note that we already have ablations on shallow models as the memory. In Table 3 (5th row), we replace our deep memory with a linear shallow memory and the results show the effectiveness of the deep memory module. Furthermore, in Figure 6, we ablate the effect of the depth of the memory (including shallow version) on the performance of the model in different model sizes and sequence lengths. While it is interesting, exploring other architectures as the memory is out of the scope of our paper and we leave it for future studies.
>
> Also, we would be happy to clarify any unclear point about the hyperparameters. We do not extensively tune any hyerperparameter and there are only two extra hyperparameters that we use: (1) chunk size: which is 16. The smaller the chunk is the better performance we can get but with the cost of slower model. (2) segment size: in which, we follow previous studies and use 2048 as sliding window (when exists) and 512 as segment size. We however, would be happy to clarify any specific part that has not been clear.
>
>
> > While the surprise-based update is interesting, it lacks deeper theoretical analysis or guarantees on memory stability or convergence.
>
> **Response:** Please note that while we understand the reviewer's interests and curiosity about the theoretical side of memory modules, we want to clarify that none of the peer studies provide such theoretical results about the capacity of the memory or why one-shot memorization works (e.g., GLA (ICML 2023), Mamba2 (ICML 2024), Mamba (COLM 2024, best paper award), Gated DeltaNet (ICLR 2025), GSA (NeurIPS 2024), Simba (NeurIPS 2024), Longhorn (NeurIPS 2024), TTT (ICML 2025, spotlight), etc.). These are indeed interesting questions that one can ask and explore in future studies. Our submission is a 9-page research paper that aims to provide **enough** evidence for its usefulness and effect. Even the initial Transformers paper, after 8 years, is still being studied, improved, and evaluated. Therefore, we believe there are indeed many interesting questions that one can study in future work.
>
>
>
> > The treatment of related work is extensive but might benefit from a clearer contrast with specific prior methods, especially regarding what exactly distinguishes Titans from prior hybrid models.
>
> **Response:** We kindly refer the reviewer to Appendix F, in which we directly compare our architectural design with most close studies, including test time training models. Also, for hybrid models, we dedicate the entire Appendix B2 to this. Similarly, we extensively discuss all TTT and fast-weight programs study in a separate sub-section (Appendix B3). However, we would be happy to provide more details if there is a specific part that the reviewer believes require more discussion.
>
>
> > The design space for combining memory and attention is still broad, and the paper only explores three specific configurations.
>
> **Response:** We agree with the reviewer that there are a broad range of ideas that one use to combine memory and attention in different configurations. Please note that this is a 9-page research paper, and we cannot discuss all possible cases. We hope that our work motivates future studies to further study the combination of memory modules and attention.
>
>
> > Can you clarify how the surprise-based memory update behaves over long sequences? It's unclear how the model avoids drift or forgetting. Some visualization or stability analysis would help.
>
> **Response:** Please note that defiantly forgetting and drift is possible over long sequences as the memory has a fixed size. This has always been a limitations for all recurrent models. (In our RULER benchmark you can see that with increasing the sequence length, the performance drops). However, in several experiments, we show that our memory module is more robust to forgetting and drift (Please see RULER results).
>
>
>
> > The role of persistent memory is vague. How is it trained, and how sensitive is the model to its size? More detail would clarify whether it's essential or just another form of prefix tuning.
>
> **Response:** Please note that persistent memory are just learnable tokens (which can also be seen as a form of prefix tuning). Such tokens are proven to be effective from different perspectives, which we have discussed in Appendix C. Looking at our ablation study in Table 3, we can see that while persistent memories are improving the performance of the model, the performance gain is not very significant and so removing them does not affect the performance of the Titans that much. Therefore, it is fair to say that Titans' performance does not rely on them and so it is not sensitive to it at all. We would be happy to discuss this further in the final version of the paper.
>
>
>
> **We hope that our above responses have addressed your concerns, and we would be more than happy to answer any remaining concern.**

---

### Official Review · Reviewer_CPKN · 2025-07-02

**Clarity:** 4
**Significance:** 3
**Originality:** 3
**Rating:** 5
**Confidence:** 3

**Summary:**

The authors introduce an interesting concept for test-time memory updates that can be thought of as a learnable “long-term” memory for transformers. Four variants of this are presented in the form of Context, Layer, Gate, and Solely Memory which balance tradeoffs between efficiency and performance on different context lengths. By explaining how to implement these methods in a highly parallelizable way the authors open the door to significant scaling and efficient use.

**Questions:**

Q1. Is there a sense from any of the tasks, either by probing or qualitative testing, what data actually gets stored in practice and what gets forgotten as a result of the surprise mechanism?

Q2. Could the memory module be shared across modalities in combined multi-modal architectures?

Q3. How sensitive is Titans’ performance to the size of its long-term memory? Are there additional ablation studies that vary the memory capacity to see whether gains saturate or degrade beyond a certain point or tradeoffs that could be made depending on the tasks?

**Ethical Concerns:**

["NO or VERY MINOR ethics concerns only"]

**Final Justification:**

I believe the authors have addressed the reviewer concerns well, providing additional data where needed and clarifying points and modifications that need to be made to the final manuscript. While I believe there are some issues that are not fully addressed in the current work around potential risks, given the current scale of the experimentation I do not think they are pressing to introduce before acceptance and can be followed-up in future work.

**Limitations:**

Further discussion of the limitations of the approach could be included in the main body of the paper, for example w.r.t. scaling limits.

**Quality:**

4

**Strengths And Weaknesses:**

S1. The authors demonstrate a promising new architecture to scale to extremely long context windows with linear-time inference. Evaluations including both the needle-in-a-haystack benchmarks and the language modeling and other tasks

S2. Providing a way to update the memory weights while leaving the core model untouched is very useful. It is a strength of the method that GPU throughput can be maintained with the added benefit of the longer context window provided by the memory module.

S3. The proposal of using the two surprise metrics (past and momentary) to generate the memory update seems useful, though it may be somewhat brittle if it requires tuning of task-specific hyperparameters.

S4. The paper is very clear: it is helpful to provide the human memory lens to frame the work and then to represent the different architecture tradeoffs as implementation options (Context/Gate/Layer) with their various efficiencies. The callbacks to human memory throughout help ground the discussion.

W1. There are some downsides to allowing the memory to be updated during inference-time: for example it might expose further security, safety, and alignment issues by giving an attacker access to a computable attack vector. While these don’t take away from the results presented it would be good to address them as potential limitations.

W2. Currently the experiments scale up to 1.3B parameter models, which show promising results. Further experimentation would be needed to validate whether this will continue to scale or if other issues might arise at larger 8B+ scale.

---

> ### Author Rebuttal · Authors · 2025-07-31
>
> **We thank the reviewer for their time and valuable comments.** We respond to your comments below:
>
> > There are some downsides to allowing the memory to be updated during inference-time: for example it might expose further security, safety, and alignment issues by giving an attacker access to a computable attack vector. While these don’t take away from the results presented it would be good to address them as potential limitations.
>
> **Response**: Thank you very much for mentioning this. We understand the importance of potential safety and alignment risks and it would be a very important direction to understand such risks in the future studies. Following your suggestion, we will make sure to discuss this as the potential limitations and advocate for further study in the future in this direction.
>
>
> > Currently the experiments scale up to 1.3B parameter models, which show promising results. Further experimentation would be needed to validate whether this will continue to scale or if other issues might arise at larger 8B+ scale.
>
> **Response**: Thank you for your comment. We understand the importance of scaling to larger scales. However, please note that training large models is extremely costly and challenging. Therefore, most academic research papers (such as Gated DeltaNet (ICLR 2025), Longhorn (NeurIPS 2024), TTT (ICML 2025), etc.) in this area focus on similar size models as ours. We followed the literature and reported the results on these scales.
>
>
>
> > Is there a sense from any of the tasks, either by probing or qualitative testing, what data actually gets stored in practice and what gets forgotten as a result of the surprise mechanism?
>
> **Response**: Please note that as an illustrative example, we have already provided Figure 10 in Appendix that shows the surprise value of different tokens in a specific prompt from the FineWeb dataset. In this example, we can see that: the surprise metric is effective to capture important tokens that are needed for the context. Words such as “Attention!” are getting higher scores, while words such as “Currently,” “Only”, etc., are mainly ignored and forgotten.
>
>
> > Could the memory module be shared across modalities in combined multi-modal architectures?
>
> **Response**: Thank you for asking this very interesting question. In this work our focus has been on the language modeling tasks and so there has been only a single modality. However, it would be a very interesting future direction to design multi-modal Titans. In theory, the memory can be shared across different modalities as the memory only memorizes tokens. If different modalities are already tokenized, then the memory module should be able to learn what tokens are more important from data (and so can be shared across different modalities).
>
>
>
> > How sensitive is Titans’ performance to the size of its long-term memory? Are there additional ablation studies that vary the memory capacity to see whether gains saturate or degrade beyond a certain point or tradeoffs that could be made depending on the tasks?
>
> **Response**: Please note that in all variants of MAC, MAG and MAL the long-term memory size is tied to the number of parameters of the model (given the key and value features, d, the memory size is d^2). Therefore, increasing or decreasing the memory size will result in #parameter change in the entire architecture. However, in Figure 6, we have ablated the performance of the memory module on different sequence lengths and memory sizes (proportional to memory size). Based on the results across model sizes, we can see that increasing the capacity of the memory enhances its effectiveness for longer context length as the memory has more parameters to store the context in. On the other hand, in each size of memory, we can see that deeper memory modules perform better at longer context length, which further validate the importance of deeper memory. Furthermore, as you might have noticed, for each model size (please see Figure 6 (a) and (b)), increasing the context beyond a point can damage the performance. The reason for this is that memory does not have the capacity to store and compress all necessary tokens. Again, we can see that deeper memories are more robust to this pattern.
>
> **We hope that our above responses have addressed your concerns, and we would be more than happy to answer any remaining concern.**

---

> > ### Comment · Reviewer_CPKN · 2025-08-06
> >
> > Thank you to the authors for preparing thorough responses to the issues I raised and those of the other reviewers. I appreciate the clarifications you provided on safety risks, scaling constraints, and the illustrative example of individual memories. I believe adding explicit discussion of potential risks that may arise from update-time memory writes addresses my main concern about attack surfaces, and I agree this belongs prominently in the final draft. Likewise, I understand the practical challenges of training 8 B-parameter models. Pointing to comparable recent work (e.g., Gated DeltaNet, Longhorn, TTT) convincingly situates your scale choices, and the ablations along with the qualitative example in Fig. 10 collectively strengthen the empirical narrative. Finally, your explanation that tokenized, modality-agnostic memory could extend to multi-modal Titans is an intriguing avenue for future exploration.
> >
> > I will plan to keep my current score.

---

### Official Review · Reviewer_Uxn4 · 2025-07-03

**Clarity:** 3
**Significance:** 4
**Originality:** 4
**Rating:** 5
**Confidence:** 4

**Summary:**

This paper proposes the novel architecture for sequence models called Titans. Titans contain test-time trainable neural long-term memory, which allows the model to surpass other sequence models like transformers and RNNs, especially on tasks with longer sequence lengths. Authors describe the mathematical framework they use to train such models, provide a comprehensive study on the architecture design, and compare their approach with a wide range of baselines on several benchmarks like language modeling, reasoning, and long-range tasks.

**Questions:**

- How do we control the number of retrieved memories (N_l)? Our memory module maps hidden states of each token to some recalled value, so the number of memories equals the number of tokens. Do we cut the last few vectors (256 in this case)? How do we handle the case when the input is shorter than 1 chunk (which is 2k if I understood correctly?)
- Please clarify your chunk-segmentation procedure. What is the size of the chunk? Is it equal to the size of the segment in MAC? Does your memory retrieve 256 tokens only for MAC, while for MAG and MAL you basically retrieve the number of vectors equal to the chunk size (= number of input tokens at the current step)?
- Could authors add comparison of results of ARMT and Titans on BABILong and discuss key differences between those two methods?
- Why don’t we just use sliding window attention in MAC just like in MAG? It is possible to keep the attention on the last tokens of the previous segment.


Typos:
- L94: date->data
- L1111: though -> through

**Ethical Concerns:**

["NO or VERY MINOR ethics concerns only"]

**Final Justification:**

The authors presented complete results for BABILong and an additional MAC+MAL variant. This increased transparency and addressed previous concerns about these results. The authors also provided the requested details during rebuttal and discussed key architectural differences between ARMT and Titans. Based on this, I've raised the quality to 4, and I hope to see updates in the camera-ready version.

**Limitations:**

yes

**Quality:**

4

**Strengths And Weaknesses:**

Strengths:
- This work proposes a unique approach of thinking about memory as an explicit test-time learner of associations between keys and values. It offers an elegant general solution to the majority of sequence tasks by delegating some computation and training to the test time, making a model to learn more general patterns during training.
- In this work, the authors comprehensively explore the capabilities of the proposed architecture on a diverse set of tasks.
- Authors provide a reasonable ablation study, showing the importance of their architectural decisions.

Weaknesses:
- It’s unclear in MAC description: how we control the number of retrieved memories (N_l). Titans memory module maps hidden states of each token to some recalled value, so the number of memories equals the number of tokens. Maybe the authors mean that we cut the last few vectors (256 in this case), but it is not explicitly stated. It’s unclear how we handle the case when the input is shorter than 1 chunk (which is 2k).
- The majority of used common-sense tasks are short (Table 1), and Titans model is basically full-attention + memory module (because sequence fits within one segment). So these tasks don’t cover sliding-window or segment-wise text processing, which is essential for long contexts. For wikitext, which requires more than 1 segment = chunk (4k tokens = 2 chunks?) MAG and MAL surpass the MAC architecture. I believe you need more comprehensive investigation on what architecture is best, especially on real-world long-context tasks (it’s unclear what long-context tasks are mentioned in Table 3).
- Results provided on BABILong on figure 8, do not include results of approaches that perform better than Titans. For example, the paper authors refer, “Associative Recurrent Memory Transformer,” as well as the very BABILong paper, includes ARMT results on BABILong, which are higher than Titan's.

---

> ### Author Rebuttal · Authors · 2025-07-31
>
> **We thank the reviewer for their time and valuable comments.** We respond to your comments below:
>
> > It’s unclear in MAC description: how we control the number of retrieved memories (N_l).
>
> **Response**: Thank you for bringing this to our attention. As you mentioned, we fix the number of memory tokens in the context and consider the last N_l tokens. When input is shorter, we pad the sequence until it matches the length of the segments/chunks.
>
> Following your suggestion, we will add these details to the final version of the paper and will make sure to properly discuss it.
>
> > The majority of used common-sense tasks are short (Table 1), and Titans model is basically full-attention + memory module (because sequence fits within one segment)
>
> **Response:** Thank you for mentioning this. Please note that we use 512 tokens as our segment size. We will make sure to make this clear in the final version of the paper. Also, please note that even our attention-free model (Titans (LMM)) outperforms all the baselines including hybrid models. Therefore, we believe these results show the strengths of Titans memory module. While we also agree with the reviewer that better understanding of the advantages of each architecture and different methods that one can use to combine attention with memory modules requires further evaluations, we want to kindly bring to your consideration that this has not been the main focus of our submission. For our experimental setups, we aimed to follow the literature (including those with hybrid architectures) and so used the same common-sense reasoning and language modeling tasks that were previously used in similar studies. However, to fully address your concerns, if you believe it is necessary, we would be happy to evaluate the performance of Titans on any downstream tasks.
>
> Thank you very much again for this valuable comment.
>
>
> >  Results provided on BABILong on figure 8
>
> **Response:** Thank you for bringing this to our attention. Please note that our main goal has been to compare sequence modeling backbones as the choice of different architectures can be orthogonal to this direction. That is, ARMT uses delta-rule to update its associative memory and so it can be replaced by our Titans (LMM) module, further enhancing the performance of ARMT. However, to fully address your concern, we provide the results of ARMT and compare it with Titans:
>
> | Model   | 2K | 4K | 8K | 16K | 32K | 64K | 128K  | 512K | 1M |
> | - | - | - | - | - | - | - | - | - | - |
> | Titans | 99 | 99  | 99 | 97  | 98  | 96  | 98  | 95  | 94 |
> | ARMT |  98 |  98 | 98  | 98  | 98  | 98  | 97  |  95  | 93 |
>
>
> On average, Titans provide a better accuracy than ARMT.
>
>
> > Please clarify your chunk-segmentation procedure.
>
> **Response:** Please note that we use chunk and segment for two different concepts. Chunks are subsequence that we use to accelerate the training process of the memory and our chunk size is 16. In fact, for each chunk, we take the gradient with respect to the last state of the previous chunk. On the other hand, we use segment to refer to the larger subsequence that we perform attention on. We will make sure to make this clear in the final version of the paper.
>
> Also, regarding the different architectures, yes, you are right. For MAG and MAL, we have the same number of outputs as the input size. While on MAC, we use a fix size memory tokens. We will make sure to discuss this in the final version of the paper.
>
> > Why don’t we just use sliding window attention in MAC just like in MAG? It is possible to keep the attention on the last tokens of the previous segment.
>
> **Response:** Thank you for mentioning this. That is totally possible and is a very interesting extension for the MAC architecture and a promising direction for future work. We will make sure to discuss it in the final paper.
>
> > Typos:
>
> **Response:** Thank you very much for bringing this to our attention. We will make sure to proofread the paper and fix all typos.
>
> **We hope that our above responses have addressed your concerns, and we would be more than happy to answer any remaining concern.**

---

> > ### Comment · Reviewer_Uxn4 · 2025-08-05
> >
> > Thank you for the detailed response and for clarifying details, terminology, and ARMT results.
> >
> > One point still requires more attention:
> >
> > The paper reports Titans results on BabiLong sequences up to 10M tokens, yet the rebuttal table compares Titans and ARMT only to 1M. The original BABILong paper includes ARMT scores for the range up to 10M, and those numbers indicate ARMT may outperform Titans at the very longest lengths.
> >
> > Could you please:
> > - Provide Titans and ARMT accuracies for all lengths (2K -> 10M) and include them in the final version.
> > - Discuss how architectural differences between ARMT (segment-lvl delta-rule associative memory) and Titans might account for the relative performance trends across length scales. In particular, which design choices, e.g., memory-update rule, segment strategy, do you believe account for Titans' strengths at some lengths and ARMT's at others?

---

> > > ### Author Response · Authors · 2025-08-07
> > >
> > > We thank the reviewer for their time, valuable comments, and engaging with us in the author-reviewer discussion period. Also, apologize for missing the last column.
> > >
> > > Following your suggestion and also to motivate investigating more architectural innovations in the future, we provide the results of full BABILong benchmark on Titans, a modified version of Titans, and ARMT as follows.
> > >
> > > | Model | 2K |4K|8K|16K|32K|64K|128K|512K|1M|10M|
> > > |-|-|-|-|-|-|-|-|-|-|-|
> > > Titans| 99| 99|99|97 | 98	|96	|98	|95	|94| 69 |
> > > Titans (GPT2 + MAC + MAL) | 99| 99|98|97 | 98	|96	|96	|95	|94| 79 |
> > > ARMT | 98|98|98|98|98|98|97|95|93|77
> > >
> > > As we discussed and also mentioned by the reviewer, the way ARMT and Titans combine their memory and attention modules are different and can be considered orthogonal since their methodology can be combined to potentially design a more powerful architecture. In fact, ARMT uses a segment-level delta-rule associative memory in a layer-wise manner while also allocating memory tokens for each segment that compress the past information. On the other hand, MAC uses a deep memory to generate memory tokens and does not take advantage of a layer-wise memory + segment attention. As supported by our empirical studies and intuitively due to the deep architecture, our neural memory module potentially provides a more expressive memory update, but on the other hand the segment-level update rule + layer-wise combination of memory and Transformers (as as done in ARMT) potentially provides more robustness in longer context lengths. As an evidence of this, we also considered a modified version, where we use both layer-wise + memory tokens, which improves the performance of Titans in 10M context length. In future work, it would indeed an interesting study to combine the best components in each of Titans and ARMT to further enhance the long-context understanding of models.
> > >
> > > Following your suggestion:
> > > - We will add these results of ARMT (2K --> 10M) in the final version of the paper,
> > > - We will discuss how architectural differences between ARMT and Titans might account for the performance in the BABILong benchmark.
> > >
> > > We hope that our above response has addressed your concerns. We would be happy to clarify any remaining point or discuss any remaining concerns.

---

> > > > ### Comment · Reviewer_Uxn4 · 2025-08-08
> > > >
> > > > Thanks for providing the full results. I appreciate the transparency and the additional MAC+MAL variant. I hope the authors will clearly separate Titans from the hybrid (MAC+MAL) variant, and support this distinction with discussion from the rebuttal as well as update text with the details from the previous comments.

---

### Comment · Area_Chair_3gmy · 2025-08-04

Dear reviewers,

Thank you for your valuable time and your expertise in reviewing. Engaging with authors is really important, and allows both authors and reviewers to gain deeper understanding of cutting edge topics. This is a unique opportunity of interaction in our community.

The author rebuttal phase is about to close, and we kindly request your prompt attention to ensure a thorough discussion.

The discussion period **ends in less than 3 days** (on Aug. 6, 11:59pm AOE ). To maintain the review timeline, we ask that you:

- Review the rebuttals,

- Engage in any ongoing discussion with fellow reviewers/authors (if applicable),

- Finalize your assessment.

If you have already completed this step, we sincerely appreciate your efforts.

Thank you for your collaboration!

Best regards,

AC

---

### Decision · Program_Chairs · 2025-09-17

**Decision:**

Accept (poster)

**Comment:**

The paper at hand proposes a new intriguing perspective on recurrent memory and new connections between RNNs and attention. All reviewers agree that the paper deserves to be accepted, though some concerns were raised regarding additional experimental results and potential directions to improve clarity.

I suggest acceptance, though I recommend that the authors work towards improving clarity in their work, particularly in terms of transitioning from "the idea" to "the implementation".